# 🎤 DuQuant: Distributing Outliers via Dual Transformation Makes Stronger Quantized LLMs

**Haokun Lin**[*1,3,4]**, Haobo Xu**[*2]**, Yichen Wu**[*4]**, Jingzhi Cui**[2]**, Yingtao Zhang**[2]**,**
**Linzhan Mou**[5]**, Linqi Song**[4]**, Zhenan Sun**[†1,3]**, Ying Wei**[†4,5]

[1] School of Artificial Intelligence, University of Chinese Academy of Sciences
[2] Tsinghua University      [3] NLPR & MAIS, Institute of Automation, CAS
[4] City University of Hong Kong      [5] Zhejiang University

haokun.lin@cripac.ia.ac.cn   xuhb20@mails.tsinghua.edu.cn
wuyichen.am97@gmail.com   znsun@nlpr.ia.ac.cn   ying.wei@zju.edu.cn

## Abstract

Quantization of large language models (LLMs) faces significant challenges, particularly due to the presence of outlier activations that impede efficient low-bit representation. Traditional approaches predominantly address *Normal Outliers*, which are activations across all tokens with relatively large magnitudes. However, these methods struggle with smoothing *Massive Outliers* that display significantly larger values, which leads to significant performance degradation in low-bit quantization. In this paper, we introduce DuQuant, a novel approach that utilizes rotation and permutation transformations to more effectively mitigate both massive and normal outliers. First, DuQuant starts by constructing the rotation matrix, using specific outlier dimensions as prior knowledge, to redistribute outliers to adjacent channels by block-wise rotation. Second, We further employ a zigzag permutation to balance the distribution of outliers across blocks, thereby reducing block-wise variance. A subsequent rotation further smooths the activation landscape, enhancing model performance. DuQuant simplifies the quantization process and excels in managing outliers, outperforming the state-of-the-art baselines across various sizes and types of LLMs on multiple tasks, even with 4-bit weight-activation quantization. Our code is available at `https://github.com/Hsu1023/DuQuant`.

## 1 Introduction

Large language models (LLMs) [51, 7, 50] have demonstrated exceptional performance across a wide range of natural language processing tasks. However, their billions of parameters present considerable deployment challenges on resource-constrained edge devices, particularly in terms of memory usage and inference speed [22, 15, 56]. In response to these challenges, network quantization methods [20, 23] have been extensively explored to minimize memory usage by converting floating-point parameters into low-bit formats [18, 32, 8], and to expedite inference by quantizing both activations and weights for accelerating the matrix multiplication process [64, 34, 74].

Among LLM quantization methods, a primary issue is the presence of activation outliers, which enlarge the quantization step sizes and subsequently cause significant accuracy loss [59]. To mitigate this problem, current research has developed various methods to address **Normal Outliers** in activations, which are *persistent in several channels across all tokens* [13, 64]. However, besides Normal Outliers, there exists another type of activation outlier [48, 35], termed **Massive Outliers**.

---

[*]Equal contribution.
[†]Corresponding authors.

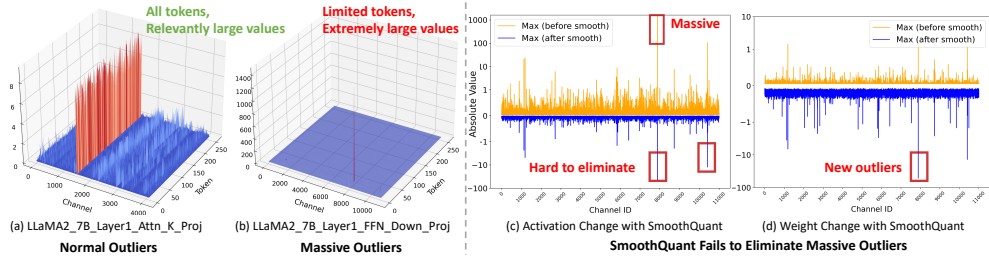

Figure 1: Visualizations of Outliers in LLaMA2-7B. (a) Input activation of Layer1 attention key projection shows Normal Outliers with relatively high magnitudes across all token sequences. (b) Input activation of Layer1 FFN down projection reveals Massive Outliers, presenting extremely high magnitudes (around 1400) at very few tokens. (c) Application of SmoothQuant on FFN down projection, illustrating its struggle with massive outliers in the Activation matrix. (d) Corresponding weight changes with SmoothQuant, highlighting the emergence of new outliers.

These outliers are characterized by their *exceedingly high values and limited occurrence in a subset of tokens*, as depicted in Figure 1(b). Unfortunately, existing LLM quantization methods struggle to effectively address these Massive Outliers. For instance, SmoothQuant [64], despite using a smooth factor to shift some of the activation outliers to the weight part, still cannot effectively handle Massive Outliers with extremely large values, as shown in Figure 1(c)(d). OmniQuant [47] and AffineQuant [39], on the other hand, exhibit training instability issues [34] due to the presence of Massive Outliers. Consequently, there is a pressing need for an LLM quantization approach that effectively addresses both Normal and Massive Outliers.

To tackle this challenge, we propose the **Du**al transformations **Quant**ization (DuQuant) method. Our motivation is to **redistribute the activation outlier values across different channels**, facilitating easier quantization. Specifically, we construct the orthogonal rotation matrix and the orthogonal permutation matrix. By multiplying these matrices with the activations, we can effectively perform column transformations on the activations, which in turn allows for the redistribution of outliers. For the **rotation transformation** aspect, we first identify specific dimensions of outliers as the prior knowledge and employ a greedy algorithm to construct the rotation matrix. To enhance the multiplication efficiency, we utilize diagonal block-wise rotation matrices, with each matrix responsible for a small portion of the activations. However, this approach may result in uneven outlier magnitudes across different blocks. Therefore, we propose the **zigzag permutation** for reordering the activation channels, which promotes a more uniform distribution across different blocks. Concretely, we distribute the channels with the highest activations across the blocks in a back-and-forth pattern. After establishing blocks with uniformly distributed outlier magnitudes, we employ another rotation transformation to further redistribute the outliers within each block. Note that we multiply the weight matrix with the transpose of the rotation and permutation matrices at the same time, preserving the linear layer equivalence and smoothing weights. Theoretical analysis confirms that the rotation and permutation transformations greatly mitigate quantization challenges induced by outliers.

As a result, DuQuant offers several clear advantages over QuaRot [2]: (1) DuQuant's optimal rotation matrix, derived through a greedy search guided by prior knowledge, surpasses QuaRot's Hadamard rotation in managing outliers; (2) our unique zigzag permutation significantly reduces activation variance across blocks, providing a distinct advantage for handling massive outliers; and (3) by jointly smoothing weights and activations, DuQuant avoids time-consuming GPTQ algorithm in QuaRot. Extensive evaluations demonstrate that our DuQuant approach significantly outperforms existing 4-bit weight-activation quantization baselines across various benchmarks. Notably, DuQuant achieves a 5% improvement in Commonsense QA tasks across all LLaMA model sizes and a 10% increase in zero-shot MMLU benchmarks for the Vicuna-v1.5-13B. Moreover, in practical applications with the LLaMA2-7B model, DuQuant not only accelerates pre-filling phase by up to 2.08× but also reduces memory usage during decoding phase by 3.50×, with minimal impact on performance: only a 0.61 increase in perplexity and a 2.71% drop in accuracy compared to the FP16 model. These results highlight the effectiveness of DuQuant in enhancing the efficiency and capacity of quantized LLMs.

## 2 Motivation

**Normal Outliers and Massive Outliers.** Previous works [13, 72, 32] have highlighted the challenge posed by activation outliers in LLMs for model compression. These outlier features consistently

manifest large values across specific feature dimensions and are present in all token sequences [64], which we refer to as *Normal Outliers*. Recently, a distinct type of outlier [48, 35], termed *Massive Outliers*, has been observed in LLMs. The primary distinctions between normal and massive outliers are: 1) Normal outliers persist across all token sequences, whereas massive outliers are confined to a limited number of tokens. 2) Massive outliers exhibit significantly larger magnitudes, often surpassing 100 and being approximately 1000 times greater than the median of other activations [48]. In our study, we delve deeper into the impact of these two distinct types of outliers on quantization.

**Massive Outliers Exist at the Second Linear Layer of FFN Module.** In contrast to previous studies [48, 35] that observe massive outliers at the output of Transformer blocks, we **first** discover that these extremely large activations exist at the input of the down-projection layer within the FFN module. As depicted in Figure 1, the input of the down-projection layer in the LLaMA2-7B model Layer 1 contains a single activation of significant magnitude (approximately 1400). This activation is isolated to one token and therefore classified as one of massive activations. This phenomenon is consistently observed across different layers and sizes of models, as illustrated in Appendix I.

**Massive Outliers Enlarge Quantization Difficulty.** Although previous studies [64, 47, 39, 1] have proposed various approaches to eliminate outlier features, they still face challenges in effectively managing massive outliers. SmoothQuant [64], for instance, attempts to shift the quantization difficulty from activations to weights by dividing the activation by a per-channel smoothing factor and multiplying it to the weight matrix. Nevertheless, we observe that this transfer at the input of the down-projection layer can cause the weights of the down-projection to display noticeable outliers, as demonstrated in Figure 1 . This issue arises because massive outliers cause the smoothing factor to become significantly large. Moreover, extremely large outliers can lead optimization-based methods to encounter problems with loss explosion. Both OmniQuant [47] and AffineQuant [39] have had to exclude their learnable parameters for the down projection layer due to unstable gradients. Given the poor accuracy observed with 4-bit quantization, QUIK [1] opts to use INT8 quantization for the down projection layer and Atom [74] applies INT8 quantization for 128 outlier channels. Consequently, massive outliers introduce new challenges to the quantization process that existing methods cannot fully address. This observation has motivated us to develop rotation and permutation transformations, which effectively handles both massive and normal outliers and achieves state-of-the-art performance.

## 3 Method

In this section, we delve into the distribution of outliers and introduce our proposed DuQuant method. The DuQuant method is built on two key components: 1) the block-diagonal rotation matrix, tasked with the local redistribution of feature outliers, and 2) the zigzag permutation, responsible for the global reordering of outliers across different blocks.

### 3.1 Preliminaries

As the common modules within each transformer block of LLMs, both Multi-head Self-Attention (MSA) and Feed-Forward Network (FFN) fundamentally consist of basic linear layers, which can be represented as, $\mathbf{Y} = \mathbf{X} \cdot \mathbf{W} \in \mathbb{R}^{T \times C_{out}}$. Here, $\mathbf{X} \in \mathbb{R}^{T \times C_{in}}$ is the activation input and $\mathbf{W} \in \mathbb{R}^{C_{in} \times C_{out}}$ denotes the weight matrix. In this paper, we focus on integer uniform quantization [25] of both activation and weight, aiming to achieve better hardware support. Specifically, the $b$-bit quantization process maps the FP16 tensor $\mathbf{X}$ to low-bit integer $\mathbf{X}_q$:

$$\mathbf{X}_q = \text{clamp}\left(\left\lfloor\frac{\mathbf{X}}{\Delta}\right\rceil + z, 0, 2^b - 1\right), \text{ where } \Delta = \frac{\max(\mathbf{X}) - \min(\mathbf{X})}{2^b - 1}, z = -\left\lfloor\frac{\min(\mathbf{X})}{\Delta}\right\rceil. \quad (1)$$

The notation $\lfloor\cdot\rceil$ means the nearest rounding operation, $\Delta$ is the quantization step size and $z$ represents the zero point. Following [64, 47, 34, 39], we employ per-token quantization for activation and per-channel quantization for weight, which entails assigning different step sizes to individual tokens of activations ($\Delta_{\mathbf{X}} \in \mathbb{R}^{T \times 1}$) and different output channels of weights ($\Delta_{\mathbf{W}} \in \mathbb{R}^{1 \times C_{out}}$).

### 3.2 The proposed DuQuant Method

To address the Normal Outliers issue stated in Section 2, current quantization methods, such as SmoothQuant [64] and OmniQuant [64], usually adopt the smooth technique. Concretely, it involves the utilization of a per-channel smoothing diagonal matrix, denoted as $\mathbf{\Lambda}$, to scale the input activation and weight matrix. The adjustment allows us to rewrite the original linear layer as $\mathbf{Y} = \mathbf{X} \cdot \mathbf{W} = (\mathbf{X} \cdot$

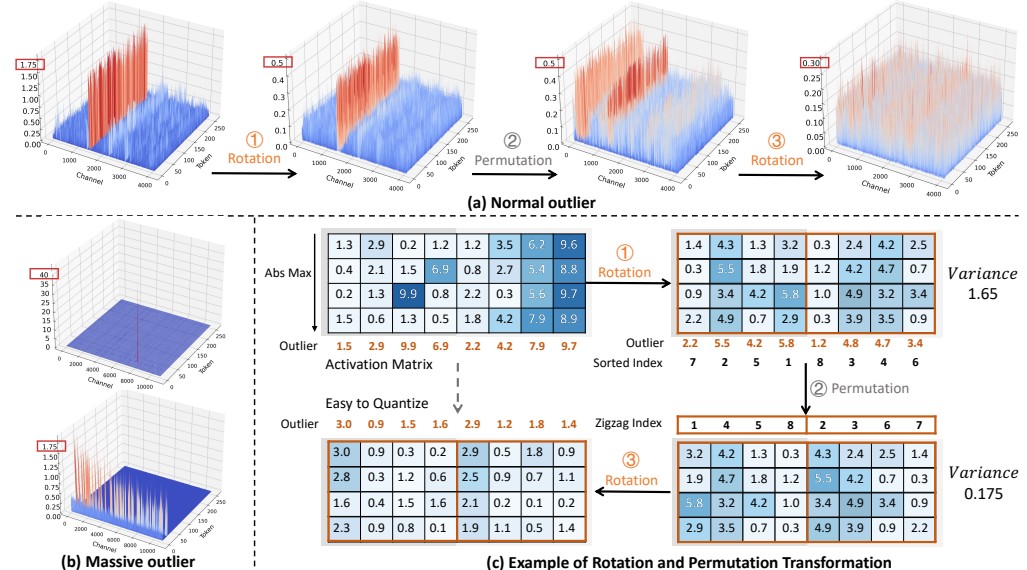

Figure 2: Transformation Steps for Activation Matrices after smooth technique. (a) Sequential transformations on Normal Outliers: ① initial rotation to reduce outliers within blocks, ② permutation to evenly distribute outliers across blocks, and ③ a second rotation for further smoothing. (b) Activation changes for Massive Outliers before and after DuQuant. (c) A sample matrix for highlighting the continual reduction of outliers through rotation and permutation, with outliers marked in dark blue.

$\mathbf{\Lambda}^{-1})(\mathbf{\Lambda} \cdot \mathbf{W})$. The diagonal element $\mathbf{\Lambda}_j$ within $\mathbf{\Lambda}$ is computed as $\mathbf{\Lambda}_j = \max(|\mathbf{X}_j|)^{\alpha}/\max(|\mathbf{W}_j|)^{1-\alpha}$, where $\alpha$ is a hyper-parameter representing the migration strength. However, despite the ability of this smoothing technique to shift the quantization challenge from activations to weights, it still faces difficulties in effectively managing Massive Outliers, as depicted in Figure 1. This challenge stems from the extremely large massive outliers inducing large scaling factors $\mathbf{\Lambda}_j$, which in turn introduce new outliers in the weight matrix and result in significant performance declines in 4-bit quantization.

According to these findings, we propose the DuQuant method, which includes the Rotation and Permutation transformations based on the smooth technique. By combining rotation transformation and channel permutation, our DuQuant method aims to redistribute these features within the activation space, thereby mitigating the effects of both Normal and Massive Outliers.

**The Rotation Transformation.** In contrast to the smooth technique, our aim is to apply a rotation matrix for row or column transformations, mitigating the impact of both Normal and Massive outliers. The ideal rotation matrix, denoted as $\mathbf{R}$, should possess the following properties: 1) $\mathbf{R}$ is an orthogonal matrix satisfying $\mathbf{R}\mathbf{R}^{\top} = \mathbf{I}$ and $|\mathbf{R}| = \pm 1$. This allows us to reformulate the linear layer within the transformer as $\mathbf{Y} = \mathbf{X} \cdot \mathbf{W} = (\mathbf{X}\mathbf{R})(\mathbf{R}^{\top}\mathbf{W})$; 2) $\mathbf{R}$ should be capable of effectively target the positions of outliers and effectively mitigating them through matrix multiplication. However, due to the Massive Outliers are usually randomly distributed within the activation space, it is challenging to directly identify the optimal rotation matrix $\mathbf{R}$ capable of mitigating outliers through a single rotation transformation. To address this problem, we employ a greedy search with prior knowledge to compute a rotation matrix $\hat{\mathbf{R}}$, thereby approximating the ideal rotation matrix $\mathbf{R}$. Specifically, the calculation of $\hat{\mathbf{R}}$ involves the following steps,

- Identify the feature dimension $d^{(1)}$ where the outlier are primarily concentrated, i.e., $d^{(1)} = \arg\max_j(\max_i |\mathbf{X}_{ij}|)$. Here, $\mathbf{X}_{ij}$ represents the element in the $i$-th row and $j$-th column of $\mathbf{X}$.

- Based on the searched dimensions $d^{(1)}$, we construct the rotation matrix as follows,

$$\mathbf{R^1} = \mathbf{E}_{d^{(1)}} \tilde{\mathbf{R}} \mathbf{Q} \mathbf{E}_{d^{(1)}}, \qquad \mathbf{Q} = \begin{bmatrix} 1 & \mathbf{O} \\ \mathbf{O} & \mathbf{Q}' \end{bmatrix}. \tag{2}$$

Here, $\mathbf{E}_{d^{(1)}}$ is the switching matrix used to swap the first and the $d^{(1)}$-th columns of the activation, and $\tilde{\mathbf{R}}$ represents an orthogonal initialized rotation matrix, in which the first row is specifically uniformly distributed. The motivation behind this is to mitigate outliers in the first column after the transformation by $\mathbf{E}_{d^{(1)}}$. To further increase the randomness of the rotation operation,

we retain the first column, where outliers have been mitigated, and randomly rotate the other columns by multiplying them with a random orthogonal matrix $\mathbf{Q}'$.

- ○ Let $N$ denote the greedy search steps, then the approximated rotation matrix $\hat{\mathbf{R}} = \mathbf{R}^1 \mathbf{R}^2 \cdots \mathbf{R}^n$, where $n = \arg\min_{k \in [1:N]} \left( \max_{i,j} |(\mathbf{X}\mathbf{R}^1 \cdots \mathbf{R}^k)_{ij}| \right)$. Each $\mathbf{R}^i$ is constructed according to Eqn. (2) and the identified feature dimension $d^{(i)}$. Appendix G provides detailed pseudo code.

Through this construction manner, we can ensure that the approximated optimal rotation matrix $\hat{\mathbf{R}}$ can effectively mitigate outliers with large magnitudes, as opposed to merely using a randomly selected orthogonal rotation matrix. Nevertheless, directly constructing the entire rotation matrix is time-consuming and results in substantial memory overhead. For fast matrix multiplication, following [63], we approximate the rotation matrix $\hat{\mathbf{R}} \in \mathbb{R}^{C_{in} \times C_{in}}$ in a block-wise manner,

$$\hat{\mathbf{R}} = \text{BlockDiag}(\hat{\mathbf{R}}_{b_1}, ..., \hat{\mathbf{R}}_{b_K}), \tag{3}$$

where $\hat{\mathbf{R}}_{b_i} \in \mathbb{R}^{2^n \times 2^n}$ denotes a square matrix of the $i$-th block, which is constructed following the three steps mentioned above. And the block numbers $K$ is calculated by $K = C_{in}/2^n$.

**The Permutation Transformation.** Despite adopting the block-diagonal rotation matrix $\hat{\mathbf{R}}$ for its time and storage efficiency, its focus on local information introduces a potential limitation in further reducing the outliers. This is because the rotation transformation, conducted within each small block, cannot integrate the information across different blocks to further minimize outliers. Consequently, one block may have relatively larger outliers while another block has smaller outliers, resulting in high variance among different blocks, as shown in Figure 2. This limitation explains that merely utilizing the block-diagonal rotation matrix is insufficient to effectively reduce the overall outliers.

To effectively mitigate the overall outliers, it is essential to balance the outliers' magnitudes among various blocks. Specifically, within each small block, we denote the largest outlier in dimension $d_j$ as $O_j$. Meanwhile, $M_{b_i}$ represents the mean value of all $O_j$ in the $i$-th block, where $i = 1, 2, ..., K$. Then the variance in activation magnitudes across various blocks can be expressed as,

$$\text{Var}([M_{b_1}, M_{b_2}, ..., M_{b_K}]). \tag{4}$$

To minimize this variance and further reduce the overall outliers, we introduce the **zigzag permutation**. Concretely, we generate a zigzag sequence that starts by assigning channels with the highest activations to the first block. The process continues by assigning channels with the next highest activations to the subsequent blocks in descending order until the end of block $K$. Upon reaching the final block, the order reverses, starting from the channel with the next highest activation and proceeding in ascending order. This back-and-forth patterning continues throughout all the blocks, ensuring that no single block consistently receives either the highest or lowest activation channels. It is worth noting that the constructed permutation is an orthogonal matrix, which we denote as $\mathbf{P}$, satisfying the conditions $\mathbf{P}\mathbf{P}^\top = \mathbf{I}$ and $|\mathbf{P}| = \pm 1$. By employing the zigzag permutation, we achieve a balanced distribution of outliers across different blocks. This allows us to use an additional rotation transformation to further smooth the outliers. Figure 2 provides an illustration of outlier mitigation.

**The Overall DuQuant Method.** To effectively mitigate both Normal and Massive Outliers, we first employ the smooth technique to shift the quantization challenge from activations to weights. Next, we introduce the block-diagonal rotation matrix $\hat{\mathbf{R}}$ to locally redistribute feature outliers within the activation space. We then propose the zigzag permutation matrix for globally balancing the outliers across different blocks, followed by another application of the block-diagonal rotation transformation. To sum up, the linear layers within the transformer can be rewrite as,

$$\mathbf{Y} = \mathbf{X} \cdot \mathbf{W} = [(\mathbf{X} \cdot \underbrace{\mathbf{\Lambda}^{-1})\hat{\mathbf{R}}_{(1)} \cdot \mathbf{P} \cdot \hat{\mathbf{R}}_{(2)}}_{\mathbf{G}}] \cdot [\underbrace{\hat{\mathbf{R}}_{(2)}^\top \cdot \mathbf{P}^\top \cdot \hat{\mathbf{R}}_{(1)}^\top (\mathbf{\Lambda} \cdot \mathbf{W})}_{\mathbf{G}^{-1}}], \tag{5}$$

where the notation $\mathbf{P}$ denotes the orthogonal permutation matrix learned via the zigzag manner, the $\hat{\mathbf{R}}_{(1)}$ and $\hat{\mathbf{R}}_{(2)}$ represent the first and second block-diagonal rotation matrix, respectively.

**Remark 1.** It is worth noting that the proposed DuQuant method can simultaneously smooth the weight matrix. While the commonly adopted smooth technique is effective, it can cause the weight matrix of the down-projection layer to exhibit pronounced outliers, leading to performance degradation. However, in the proposed DuQuant method, the rotation transformation we designed is applied to not only the activation input but also the weight matrix. As a result, the outliers induced by the smooth technique can be mitigated through our approximated rotation matrix $\hat{\mathbf{R}}$, yielding a

smoother, more quantization-friendly weight matrix. Moreover, this approach eliminates the reliance on complex weight quantization techniques, such as GPTQ [18] used in Atom [74] and QuaRot [2].

**Remark 2.** To further decrease the computation and memory costs, we initially construct the $k$-th block rotation matrix $\hat{\mathbf{R}}_{b_k}$, with the $k$-th block containing the largest outlier. We then assign $\hat{\mathbf{R}}_{b_i} = \hat{\mathbf{R}}_{b_k}$ for all $1 \leq i \leq K$. This strategy not only effectively mitigates the impact of outliers, but also reduces the number of block rotation matrices from $K$ to 1, significantly reducing computation and memory requirements. Importantly, incorporating the invertible matrix $\mathbf{G}$ from Eqn. (5) significantly eases the quantization challenges for $\mathbf{X}$ and $\mathbf{W}$. Consequently, the quantization process acts as $\mathbf{Y} = (\mathbf{XG})(\mathbf{G}^{-1}\mathbf{W}) = \hat{\mathbf{X}} \cdot \hat{\mathbf{W}} \approx \Delta_{\hat{\mathbf{X}}}\Delta_{\hat{\mathbf{W}}}(\hat{\mathbf{X}}_q - z_{\hat{\mathbf{X}}})(\hat{\mathbf{W}}_q - z_{\hat{\mathbf{W}}})$.

### 3.3 Theoretical Analysis

To further demonstrate the effectiveness of the proposed DuQuant method, we conduct a theoretical analysis of the rotation and permutation transformations. Theorem 1 shows that within each block, the constructed rotation matrix effectively mitigates the maximum outlier, thereby reducing the outlier magnitude through a greedy search. Theorem 2 reveals that the employed zigzag permutation ensures a balanced upper bound shared among different blocks. This suggests that the zigzag permutation effectively reduces the variance shown in Eqn. (4) and thus assists the rotation matrix in further decreasing the outliers. Please refer to Appendix B for detailed proofs.

**Theorem 1** (Rotation). *For the activation input* $\mathbf{X} \in \mathbb{R}^{T \times C_{in}}$, $\hat{\mathbf{R}} \in \mathbb{R}^{2^n \times 2^n}$ *is a diagonal block matrix constructed as per Eqn. (3). For a specific block* $b_i$, *let* $O_j(\cdot)$ *represent the maximum outlier of the* $j$-th *dimension* $d_j$ *within the input. Then, we can deduce that,*

$$\max_{1 \leq j \leq 2^n} O_j(\mathbf{X}_{b_i}\hat{\mathbf{R}}_{b_i}) \leq \max_{1 \leq j \leq 2^n} O_j(\mathbf{X}_{b_i}). \tag{6}$$

**Theorem 2** (Zigzag Permutation). *For the activation input* $\mathbf{X} \in \mathbb{R}^{T \times C_{in}}$, *it can be divided into* $K$ *blocks, where* $K = C_{in}/2^n$. *Let* $O_j$ *denote the max outlier of the dimension* $d_j$ *in* $\mathbf{X}$, *the reordered outliers from large to small is expressed as* $O^{(1)}, O^{(2)}, ..., O^{(C_{in})}$. *Moreover, the* $M_{b_i}$ *represents the mean value of all* $O_j$ *in the* $i$-th *block,* $i = 1, 2, ..., K$. *Let* $\delta := \max\{|O^{(i+1)} - O^{(i)}|\}, i = 1, 2, ..., C_{in}-1$. *Then, following the zigzag permutation described in Section 3.2, the mean value* $M_{b_i}$ *within each* $i$-th *block consistently satisfies,*

$$M_{b_i} \leq O^{(1)} + \frac{(2^n K - 1)(2^{n-1} - 1)}{2^n}\delta, \qquad i = 1, 2, 3, ..., K. \tag{7}$$

## 4 Experiment

**Models and Evaluations.** We apply our DuQuant on pre-trained LLMs: LLaMA (7B-65B) [51], LLaMA2 (7B-70B) [52], LLaMA3 (8B, 70B), Mistral, Phi2 and instruction-tuned LLMs: Vicuna-v1.5 (7B-13B) [10]. We evaluate quantized pre-trained LLMs on language generation tasks and commonsense QA tasks. Specifically, we assess the perplexity on WikiText2 [40] and C4 [44] datasets, as well as the zero-shot accuracy on PIQA [6], ARC [12], BoolQ [11], HellaSwag [70], and WinoGrande [45] datasets. Moreover, we evaluate quantized Vicuna models on MMLU [21] and MT-Bench [76] benchmarks, as well as their long-form generative capabilities on LongBench [4].

**Implementation Details.** In line with prior studies [34, 47, 39], we apply per-token activation quantization and per-channel weight quantization. Given that W8A8 quantization has been established as lossless in precision by SmoothQuant [64], our primary evaluation in this paper focuses on 4-bit and 6-bit quantization for weights and activations. As for details, we quantize all intermediate activations, excluding the SoftMax output. Moreover, we have developed two types of quantized models, denoted as **DuQuant** and **DuQuant+LWC**. For DuQuant, we employ round-to-nearest quantization, using a clipping ratio of 0.9 for activations and 0.8 for weights. To improve weight matrix quantization, DuQuant+LWC integrates the learnable weight clipping (LWC) technique from Omni-Quant. Concretely, LWC adjusts weights by training parameters $\gamma, \beta \in [0, 1]$ to compute step size $\Delta = \frac{\gamma \max(\mathbf{X}) - \beta \min(\mathbf{X})}{2^b - 1}$ in Eqn. (1). Notably, the smoothing diagonal matrix and the learned weight clipping factor can be integrated into the quantized weights, introducing no additional computational or memory costs. More details and hyperparameters are left in Appendix C.

**Baselines.** We compare with state-of-the-art (SOTA) weight-activation PTQ methods, including SmoothQuant [64], Outlier Supression+ [59], OmniQuant [47], QLLM [34], AffineQuant [39], and Atom [74]. For Atom, we reproduce the results with no group-wise asymmetric quantization.

Table 1: Perplexity (↓) results under 4-bit weight-activation quantization. The results for W6A6 can be found in Table D8. Atom and OmniQuant unprocessed group-query attention for LLaMA2-70B.

| Dataset | #Bit | Method | 1-7B | 1-13B | 1-30B | 1-65B | 2-7B | 2-13B | 2-70B |
|---------|------|--------|------|-------|-------|-------|------|-------|-------|
| WikiText2 | FP16 | - | 5.68 | 5.09 | 4.10 | 3.53 | 5.47 | 4.88 | 3.31 |
| | W4A4 | SmoothQuant | 25.25 | 40.05 | 192.40 | 275.53 | 83.12 | 35.88 | 26.01 |
| | | OmniQuant | 11.26 | 10.87 | 10.33 | 9.17 | 14.26 | 12.30 | NaN |
| | | AffineQuant | 10.28 | 10.32 | 9.35 | - | 12.69 | 11.45 | - |
| | | QLLM | 9.65 | 8.41 | 8.37 | 6.87 | 11.75 | 9.09 | 7.00 |
| | | Atom | 8.15 | 7.43 | 6.52 | 5.14 | 8.40 | 6.96 | NaN |
| | | **DuQuant** | 6.40 | 5.65 | 4.72 | 4.13 | 6.28 | 5.42 | 3.79 |
| | | **DuQuant+LWC** | **6.18** | **5.47** | **4.55** | **3.93** | **6.08** | **5.33** | **3.76** |
| C4 | FP16 | | 7.08 | 6.61 | 5.98 | 5.62 | 6.97 | 6.46 | 5.52 |
| | W4A4 | SmoothQuant | 32.32 | 47.18 | 122.38 | 244.35 | 77.27 | 43.19 | 34.61 |
| | | OmniQuant | 14.51 | 13.78 | 12.49 | 11.28 | 18.02 | 14.55 | NaN |
| | | AffineQuant | 13.64 | 13.44 | 11.58 | - | 15.76 | 13.97 | - |
| | | QLLM | 12.29 | 10.58 | 11.51 | 8.98 | 13.26 | 11.13 | 8.89 |
| | | Atom | 10.34 | 9.57 | 8.56 | 8.17 | 10.96 | 9.12 | NaN |
| | | **DuQuant** | 7.84 | 7.16 | 6.45 | 6.03 | 7.90 | 7.05 | 5.87 |
| | | **DuQuant+LWC** | **7.73** | **7.07** | **6.37** | **5.93** | **7.79** | **7.02** | **5.85** |

Table 2: Zero-shot QA (↑) results of LLaMA1 models under 4-bit weight-activation quantization. The results for LLaMA2 models and W6A6 quantization can be found in Table D1 D9, and D10.

| Model | Method | PIQA | ARC-E | ARC-C | BoolQ | HellaSwag | WinoGrande | Avg. |
|-------|--------|------|-------|-------|-------|-----------|------------|------|
| | FP16 | 77.47 | 52.48 | 41.46 | 73.08 | 73.00 | 67.07 | 64.09 |
| LLaMA1-7B W4A4 | SmoothQuant | 49.80 | 30.40 | 25.80 | 49.10 | 27.40 | 48.00 | 38.41 |
| | OS+ | 62.73 | 39.98 | 30.29 | 60.21 | 44.39 | 52.96 | 48.43 |
| | OmniQuant | 66.15 | 45.20 | 31.14 | 63.51 | 56.44 | 53.43 | 52.65 |
| | AffineQuant | 69.37 | 42.55 | 31.91 | 63.73 | 57.65 | 55.33 | 53.42 |
| | QLLM | 68.77 | 45.20 | 31.14 | - | 57.43 | 56.67 | 51.84 |
| | Atom | 71.44 | 47.74 | 35.49 | 67.71 | 63.89 | 55.01 | 56.88 |
| | **DuQuant** | **76.44** | **50.04** | **38.99** | **70.98** | 69.39 | **64.72** | **61.76** |
| | **DuQuant+LWC** | 76.22 | **50.04** | 38.31 | 70.09 | **69.82** | 62.59 | 61.18 |
| | FP16 | 79.10 | 59.89 | 44.45 | 68.01 | 76.21 | 70.31 | 66.33 |
| LLaMA1-13B W4A4 | SmoothQuant | 61.04 | 39.18 | 30.80 | 61.80 | 52.29 | 51.06 | 49.36 |
| | OS+ | 63.00 | 40.32 | 30.38 | 60.34 | 53.61 | 51.54 | 49.86 |
| | OmniQuant | 69.69 | 47.39 | 33.10 | 62.84 | 58.96 | 55.80 | 54.37 |
| | AffineQuant | 66.32 | 43.90 | 29.61 | 64.10 | 56.88 | 54.70 | 52.58 |
| | QLLM | 71.38 | 47.60 | 34.30 | - | 63.70 | 59.43 | 55.28 |
| | Atom | 71.38 | 49.07 | 36.69 | 64.53 | 68.00 | 58.56 | 58.04 |
| | **DuQuant** | 77.26 | **58.04** | **41.55** | **67.55** | 73.62 | **66.69** | **64.12** |
| | **DuQuant+LWC** | **77.64** | 57.32 | 41.21 | 66.79 | **74.12** | 65.98 | 63.84 |
| | FP16 | 80.08 | 58.92 | 45.47 | 68.44 | 79.21 | 72.53 | 67.44 |
| LLaMA1-30B W4A4 | SmoothQuant | 58.65 | 35.53 | 27.73 | 60.42 | 35.56 | 48.06 | 44.83 |
| | OS+ | 67.63 | 46.17 | 34.40 | 60.70 | 54.32 | 52.64 | 52.62 |
| | OmniQuant | 71.21 | 49.45 | 34.47 | 65.33 | 64.65 | 59.19 | 56.63 |
| | AffineQuant | 70.84 | 49.41 | 37.12 | 70.12 | 65.53 | 58.64 | 58.61 |
| | QLLM | 73.83 | 50.67 | 38.40 | - | 67.91 | 58.56 | 57.87 |
| | Atom | 71.98 | 49.07 | 40.02 | 66.85 | 70.45 | 58.64 | 59.50 |
| | **DuQuant** | 78.56 | **56.99** | 42.32 | 66.73 | 76.70 | 69.61 | 65.15 |
| | **DuQuant+LWC** | **78.73** | 56.52 | **43.17** | **68.84** | **77.53** | **70.96** | **65.96** |
| | FP16 | 80.79 | 58.71 | 46.24 | 82.29 | 80.72 | 77.50 | 71.04 |
| LLaMA1-65B W4A4 | SmoothQuant | 64.47 | 40.44 | 29.82 | 59.38 | 39.90 | 52.24 | 47.71 |
| | OS+ | 68.06 | 43.98 | 35.32 | 62.75 | 50.73 | 54.30 | 52.52 |
| | OmniQuant | 71.81 | 48.02 | 35.92 | 73.27 | 66.81 | 59.51 | 59.22 |
| | QLLM | 73.56 | 52.06 | 39.68 | - | 70.94 | 62.90 | 59.83 |
| | Atom | 74.48 | 51.60 | 40.61 | 73.76 | 73.78 | 62.12 | 62.73 |
| | **DuQuant** | 79.71 | 57.95 | **45.05** | **79.82** | 78.66 | **72.29** | **68.91** |
| | **DuQuant+LWC** | **79.98** | **58.29** | 44.80 | 77.89 | **79.22** | 72.21 | 68.73 |

## 4.1  Main Results

**Quantization of LLaMA1 and LLaMA2 Models.**    We conduct a comprehensive comparison of our DuQuant with several SOTA baselines on LLaMA1 and LLaMA2 models. Results for W4A4 quantization are presented in this Section, while results for W6A6 quantization are provided in

Table 3: Zero-shot and five-shot results on the MMLU benchmark for Vicuna-v1.5-13B under 4-bit weight-activation quantization. The results for Vicuna-v1.5-7b can be found in Table D2.

| Model | Method | MMLU (0 shot) ↑ | | | | | MMLU (5 shot) ↑ | | | | |
|---|---|---|---|---|---|---|---|---|---|---|---|
| | | STEM | Hums | Social | Others | Avg. | STEM | Hums | Social | Others | Avg. |
| Vicuna-v1.5-13B W4A4 | FP16 | 43.70 | 50.48 | 62.72 | 62.74 | 54.54 | 44.96 | 51.97 | 65.26 | 62.40 | 55.78 |
| | SmoothQuant | 21.70 | 24.29 | 22.13 | 23.16 | 22.82 | 25.31 | 24.97 | 26.00 | 27.08 | 25.84 |
| | OmniQuant | 26.81 | 26.57 | 30.35 | 28.75 | 28.12 | 28.79 | 27.29 | 31.13 | 28.99 | 29.05 |
| | Atom | 32.54 | 39.60 | 46.02 | 46.11 | 41.07 | 35.35 | 39.21 | 59.72 | 45.77 | 45.01 |
| | **DuQuant** | **40.82** | 46.61 | 58.73 | 57.59 | **50.94** | 40.92 | **48.78** | **60.42** | 57.71 | **51.96** |
| | **DuQuant+LWC** | 40.13 | **47.48** | **58.86** | **57.83** | **51.08** | **41..42** | 48.52 | 58.73 | **57.74** | 51.61 |

Table 4: Long-context generation results for 4-bit Vicuna models on the LongBench benchmark.

| Vicuna | Setting | Qasper | QMSum | MultiNews | TREC | TriviaQA | SAMSum | DuReader | RepoBench-P | Avg |
|---|---|---|---|---|---|---|---|---|---|---|
| Vicuna-v1.5-7B W4A4 | FP16 | 23.27 | 21.07 | 26.91 | 66.00 | 82.59 | 41.06 | 25.53 | 48.23 | 41.83 |
| | SmoothQuant | 4.11 | 2.00 | 6.05 | 15.00 | 1.62 | 1.55 | 4.24 | 25.92 | 7.56 |
| | OmniQuant | 1.62 | 3.93 | 2.64 | 1.00 | 0.81 | 0.61 | 1.87 | 14.97 | 3.43 |
| | Atom | 17.97 | 20.24 | 24.60 | 58.00 | 67.20 | 37.94 | 19.41 | 29.34 | 34.34 |
| | **DuQuant** | **19.98** | **21.15** | **25.85** | **64.00** | **78.91** | **42.24** | **23.15** | **47.66** | **40.37** |
| Vicuna-v1.5-13B W4A4 | FP16 | 24.41 | 21.24 | 26.53 | 68.00 | 86.81 | 41.97 | 27.57 | 43.08 | 42.45 |
| | SmoothQuant | 2.18 | 2.95 | 3.54 | 1.50 | 1.83 | 0.35 | 6.71 | 11.57 | 3.83 |
| | OmniQuant | 0.68 | 1.78 | 2.83 | 9.00 | 1.13 | 0.45 | 13.83 | 8.46 | 4.77 |
| | Atom | 17.67 | 20.23 | 23.39 | 59.00 | 80.75 | 38.72 | 21.79 | 37.31 | 37.36 |
| | **DuQuant** | **18.93** | **20.72** | **26.59** | **66.50** | **83.04** | **42.67** | **26.02** | **38.09** | **40.32** |

Table 5: Perplexity and QA results of LLaMA3-8B under 4-bit/6-bit weight-activation quantization.

| #Bits | Method | WikiText2↓ | C4↓ | PTB↓ | PIQA | ARC-E | ARC-C | BoolQ | HellaSwag | WinoGrande | Avg. ↑ |
|---|---|---|---|---|---|---|---|---|---|---|---|
| FP16 | - | 6.14 | 8.88 | 9.91 | 80.85 | 77.78 | 53.41 | 81.28 | 79.16 | 72.84 | 74.22 |
| LLaMA3-8B W6A6 | SmoothQuant | 7.07 | 9.57 | 11.69 | 78.94 | 75.88 | 49.49 | 77.58 | 77.39 | 70.8 | 71.68 |
| | OmniQuant | 7.24 | 9.82 | 11.90 | 78.90 | 73.95 | 47.35 | 74.95 | 76.77 | 70.56 | 70.41 |
| | AffineQuant | 7.35 | 9.99 | 12.30 | 78.73 | 73.32 | 46.08 | 74.59 | 77.08 | 70.88 | 70.11 |
| | **DuQuant** | **6.27** | **8.38** | 10.77 | **80.20** | 77.27 | 52.05 | **80.12** | **79.14** | 72.77 | 73.59 |
| | **DuQuant+LWC** | **6.27** | **8.38** | 10.78 | 79.71 | **77.57** | **53.07** | 80.00 | 78.70 | **73.09** | **73.69** |
| LLaMA3-8B W4A4 | SmoothQuant | 210.19 | 187.93 | 278.02 | 54.57 | 31.9 | 24.23 | 52.72 | 31.26 | 51.14 | 40.97 |
| | OmniQuant | 3.64e3 | 2.80e3 | 3.09e3 | 50.22 | 26.94 | 24.57 | 37.98 | 26.55 | 50.20 | 36.08 |
| | AffineQuant | 21.21e3 | 34.60e3 | 16.72e3 | 50.71 | 25.93 | 26.02 | 40.55 | 26.07 | 48.46 | 36.29 |
| | Atom | 22.14 | 31.83 | 40.04 | 62.95 | 49.45 | 30.12 | 60.31 | 53.75 | 56.04 | 52.10 |
| | **DuQuant** | 8.56 | 11.98 | 13.66 | 75.68 | 68.48 | 41.81 | 71.99 | 73.07 | 66.22 | 66.21 |
| | **DuQuant+LWC** | **8.06** | **11.29** | **13.19** | **76.22** | **70.41** | **43.69** | **74.34** | **73.87** | **67.80** | **67.72** |

Appendix D. Table 1 indicates that our DuQuant quantized models notably outperform other baselines on both the WikiText2 and C4 datasets. Notably, LWC technique further enhances model capacity, with our DuQuant+LWC achieving comparable performance with FP16 models. Table 2 and Table D1 showcase the zero-shot accuracy of W4A4 quantization on Commonsense QA tasks, where DuQuant significantly improves the average accuracy. Our method surpasses QLLM by +9%, and Atom by +5% for all model sizes. These results demonstrate the superiority of our rotation and permutation transformation, which establishes new SOTA performance by effectively eliminating outlier features.

**Quantization of Instruction-tuned Models.** We quantize Vicuna-v1.5 [10] models to assess the generalizability of our DuQuant. Table 3 illustrates that our quantized models surpass the baselines across all task categories on MMLU benchmark. For Vicuna-13B, our DuQuant+LWC surpasses Atom by 10.01% under zero-shot settings and 6.95% under five-shot settings. Moreover, we compare our DuQuant with Atom and OmniQuant using MT-Bench and utilize GPT-4 to evaluate the answers from quantized models. As shown in Figure 3, DuQuant quantized models significantly outperform both Atom and OmniQuant in win rates. Specifically, for Vicuna-7B, DuQuant only lost 16 and 1 times to Atom and OmniQuant, respectively, while achieving 68 and 155 wins against them.

**Evaluation of Long-context Generation.** To further evaluate the long-text generative capabilities, we follow [37, 33, 55] and conduct a comprehensive comparison of DuQuant against state-of-the-art baselines on the LongBench [4], which includes a variety of generative tasks to provide a broader evaluation. We set the maximum sequence length to 3500 for Vicuna models, with results presented in Table 4. DuQuant achieves performance comparable to FP16 models, demonstrating the effectiveness of our dual transformations. More detailed results on different subtasks are listed in Table D3, D4.

**Quantization of LLaMA3 Models.** LLaMA3, known for its superior performance in various tasks, faces significant degradation in low-bit quantization [24]. To address this, we apply our DuQuant to quantize LLaMA3-8B. Table 5 displays the perplexity and zero-shot accuracy results. Notably, under W6A6 setting, our DuQuant achieves performance comparable to FP16 model. Furthermore, unlike other methods that show weaker results under W4A4 setting, our DuQuant maintains competitive performance, indicating its robustness with LLaMA3. We attribute this success to the advanced handling of outliers achieved through dual transformations, which is not restricted to specific models.

## 4.2 Ablation Study

**Module-wise Impact.** We ablate four distinct operations within DuQuant: 1) only the smoothing technology like SmoothQuant; 2) one rotation following the smoothing operation; 3) a sequence of rotation, permutation, and another rotation without smoothing; and k4) full DuQuant approach. Table 6 shows that the smoothing operation plays a basic role in our DuQuant by shifting activation outliers to weight. The initial rotation significantly enhances model performance, yielding competitive PPL results. Finally, permutation combined with a second rotation further enhances the quantized model.

Table 6: Influence of different components in DuQuant under 4-bit weight-activation quantization.

| Modules | | | | LLaMA2-7B | | LLaMA2-13B | |
|---|---|---|---|---|---|---|---|
| Smooth | Rotation 1 | Permutation | Rotation 2 | WikiText2 ↓ | C4 ↓ | WikiText2 ↓ | C4 ↓ |
| ✓ | | | | NaN | 1379.46 | 160.30 | 203.87 |
| | ✓ | | | 8.48 | 10.63 | 14.32 | 21.73 |
| ✓ | ✓ | | | 7.92 | 10.64 | 5.96 | 7.94 |
| | ✓ | ✓ | ✓ | 6.79 | 8.51 | 6.06 | 8.03 |
| ✓ | ✓ | ✓ | ✓ | **6.28** | **7.90** | **5.42** | **7.05** |

**Influence of Normal/Massive Outliers.** In this section, we comprehensively explore the influence of massive and normal outliers on quantization. Notably, we observe that massive outliers primarily occur at the down-projection of the FFN module. To isolate their effect, we remove the rotation and permutation transformations, applying only the smoothing technique to all down-projection inputs. The resulting perplexity for LLaMA2-7B and LLaMA-13B showed significant degradation, presented in Table 7. Conversely, when we eliminate the rotation and permutation transformations for normal outliers, the performance decrease was noticeable but less severe compared to massive outliers. These findings indicate that: 1) massive outliers exert a more substantial impact on quantization, corroborating our claims in Section 2; 2) the smoothing technique alone struggles to fully mitigate the influence of outliers, particularly massive ones; and 3) our rotation and permutation methods prove highly effective against both types of outliers, leading to superior performance.

Table 7: Outliers impact on quantization. We only apply the smooth technique on Normal and Massive outliers for W4A4 quantization.

| Outlier Type | | LLaMA2-7B | | LLaMA2-13B | |
|---|---|---|---|---|---|
| Normal | Massive | WikiText2 ↓ | C4 ↓ | WikiText2 ↓ | C4 ↓ |
| | ✓ | 18.16 | 26.42 | 10.51 | 16.01 |
| ✓ | | 10.88 | 13.89 | 7.87 | 10.52 |
| ✓ | ✓ | **6.28** | **7.90** | **5.42** | **7.05** |

Figure 3: GPT-4 evaluation on the MT-Bench.

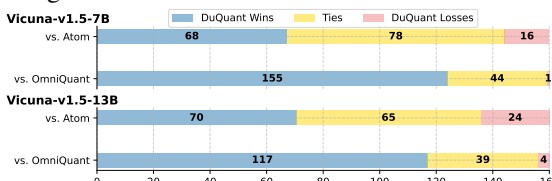

**Comparison with QuaRot [2]** In light of the recent introduction of Hadamard rotations by QuaRot[2] to eliminate outlier features, we have undertaken a detailed analysis to highlight the key differences between our DuQuant and QuaRot. To ensure a balanced evaluation, we have re-implemented QuaRot in accordance with our quantization settings. The results demonstrate that 1) the rotation matrix constructed by DuQuant outperforms QuaRot's approach of simply selecting a randomly initialized Hadamard matrix. As depicted in Figure 10, our DuQuant more effectively smooths activations than QuaRot. This is attributed to the prior knowledge utilized by DuQuant to accurately target the outliers; 2) As demonstrated by the perplexity in Table 8, QuaRot employs GPTQ for their weight quantization method, whereas our DuQuant, with its sophisticated outlier management, attains competitive results using RTN quantization. The superiority proves the effectiveness of our zigzag permutation to enhance capacity. For a more comprehensive comparison, please refer to Appendix F.

Table 8: PPL (↓) comparison under W4A4 setting.

| Method | 1-7B | 1-13B | 1-30B | 2-7B | 2-13B |
|---|---|---|---|---|---|
| FP16 | 5.68 | 5.09 | 4.10 | 5.47 | 4.88 |
| QuaRot-RTN | 7.08 | 6.57 | 5.44 | 9.66 | 6.73 |
| QuaRot-GPTQ | 6.44 | 5.63 | 4.73 | 6.39 | 5.75 |
| **DuQuant** | 6.40 | 5.65 | 4.72 | 6.28 | 5.42 |
| **DuQuant**_+LWC_ | **6.18** | **5.47** | **4.55** | **6.08** | **5.33** |

Figure 4: LLaMA2-7B Attention key_proj.

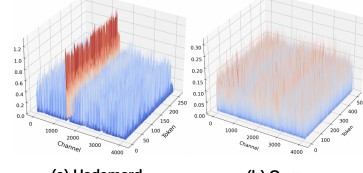

(a) Hadamard      (b) Ours

**Permutation Frequency.** We conduct ablations on the rotation and permutation frequencies in DuQuant. As shown in Figure 5, "Perm 1" (two rotations with one permutation) achieves stronger

performance compared with "Perm 0" (no permutation), while incurring an additional 8.9% computational cost on LLaMA2-7B and 9.3% on LLaMA2-13B compared to the W4A4 setup. Considering the approximately $2\times$ speedup and the impressive performance, these additional costs are deemed acceptable. Further permutations, like "Perm 2," do not improve performance and reduce inference efficiency. Consequently, "Perm 1" strikes the best balance between perplexity and inference speed, making it the optimal configuration for DuQuant.

**Inference Speedup.** To assess the inference speedup delivered by our DuQuant, we adopt the measurement strategy and W4A4 kernel from [2]. We evaluate the layer-wise speedup of LLaMA2 models on one NVIDIA 3090 GPU, with results detailed in Table 9 and 10. We set the pre-filling sequence length at 2048 and decode for 128 steps. In the pre-filling stage, DuQuant achieves a $2.08\times$ speedup over FP16 for LLaMA2-7B and a $2.34\times$ speedup for LLaMA2-13B, with slight variations across different batch sizes. In the decoding stage, batching the token generation phase yields high throughput without any downside [43]. Consequently, we enlarge the batch size to 64 and the results for LLaMA2-7B in Table 10 prove DuQuant achieves speedup comparable to QuaRot. More detailed analyses and end-to-end speedup are available in Appendix E.1.

Table 9: Layer-wise speedup during pre-filling stage for 4-bit weight-activation quantization.

| Model | Batch Size | Speedup |
|---|---|---|
| | 1 | $1.95\times$ |
| LLaMA2-7B | 4 | $2.03\times$ |
| | 16 | $2.08\times$ |
| | 1 | $2.15\times$ |
| LLaMA2-13B | 4 | $2.30\times$ |
| | 16 | $2.34\times$ |

Figure 5: Computational overhead analysis.

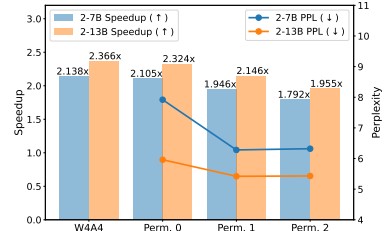

**Memory Consumption.** We measure the peak memory usage of DuQuant with the W4A4 kernel on LLaMA2-7B using a single NVIDIA 3090 GPU. We process 2048 tokens for pre-filing and run 128 decoding steps, with the results listed in Table 11. In the pre-filling stage, DuQuant, SmoothQuant, and QuaRot achieve up to $3.2\times$ memory reduction, while QLLM performs worse. In the decoding stage, DuQuant maintains strong memory efficiency, with superior performance.

Table 10: Decoding stage.

| INT4, BS=64 | Speedup |
|---|---|
| FP16 | - |
| SmoothQuant | $1.508\times$ |
| QLLM | OOM |
| QuaRot | $1.442\times$ |
| DuQuant | $1.321\times$ |

Table 11: Peak memory usage with a batch size of 1.

| LLaMA2-7B | pre-filling (GB) | Saving | Decoding (GB) | Saving |
|---|---|---|---|---|
| FP16 | 15.282 | - | 13.638 | - |
| SmoothQuant | 4.782 | $3.196\times$ | 3.890 | $3.506\times$ |
| QLLM | 5.349 | $2.857\times$ | 3.894 | $3.502\times$ |
| QuaRot | 4.784 | $3.194\times$ | 3.891 | $3.505\times$ |
| DuQuant | 4.786 | $3.193\times$ | 3.893 | $3.503\times$ |

**Runtime.** Our DuQuant stands out for its efficiency, surpassing other baselines [47, 34, 39, 74]. The block-wise rotation ensures fast multiplication between the rotation and activation matrices. Zigzag permutation,

Table 12: Quantization runtime on one NVIDIA A100.

| Model | Omni. | Affine. | QLLM | Atom | DuQuant |
|---|---|---|---|---|---|
| LLaMA2-7B | 2.0h | 9.1h | 1.1h | 20min | 50s |
| LLaMA2-13B | 3.2h | 16.0h | 1.7h | 36min | 71s |
| LLaMA2-70B | 14.6h | 18.6h | 9.3h | 3.5h | 270s |

involving simple channel swaps, is much faster than complex algorithms like Simulated Annealing, as discussed in Appendix E.3. Moreover, the advanced management of outliers makes DuQuant not rely on GPTQ or gradient-based training. Hence, DuQuant enables a rapid quantization process shown in Table F24, e.g., we successfully quantize LLaMA2-13B in just 71s with superior performance.

## 5 Conclusion

In conclusion, this paper presents DuQuant, an innovative quantization strategy for large language models (LLMs) that effectively addresses the challenge of outlier activations. By integrating rotation and permutation transformations, DuQuant effectively mitigates the impacts of both massive and normal outliers. This strategic redistribution of outliers not only simplifies the quantization process but also leads to substantial improvements in model performance. Consequently, DuQuant establishes new state-of-the-art results in 4-bit weight-activation quantization scenarios. This advancement enhances the deployment of efficient LLMs in resource-constrained environments.

## Acknowledgements

This work is supported in part by the National Natural Science Foundation of China (Grant No. U23B2054, 62276263, and 62371411), the Research Grants Council of the Hong Kong SAR under Grant GRF 11217823 and Collaborative Research Fund C1042-23GF, InnoHK initiative, the Government of the HKSAR, Laboratory for AI-Powered Financial Technologies.

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

# Appendix Overview

# A  Related Work

## A.1  Network Quantization

Neural networks achieve great success in recent years [65, 77, 38, 61, 31, 19, 62], quantization [1, 63, 67, 74, 57, 75] is a widely utilized technique aimed at reducing model size and memory usage. Research in this area generally falls into two main categories: quantization-aware training (QAT) [71, 3, 49] and post-training quantization (PTQ) [41, 60, 73]. QAT involves training quantized model weights using additional data, often with the assistance of a straight-through estimator (STE) [5]. However, the computational cost associated with QAT poses challenges, particularly for large language models (LLMs) with millions of parameters, which necessitate significant amounts of data for retraining [36, 16, 9]. In contrast, PTQ has gained popularity for LLMs [54, 22, 69, 58, 37] due to its efficient approach, involving the training of quantized models using a small amount of data, known as calibration data [18]. However, PTQ often leads to significant performance degradation, especially when employing low-bit settings [18, 53, 46, 30]. Consequently, our work focuses on enhancing the performance of low-bit PTQ quantized models.

## A.2  Post Training Quantization of LLM

Post-training quantization for LLMs can be categorized into weight-only quantization [32, 14, 28, 29] and weight-activation quantization [68, 64, 74]. We focus on 4-bit weight-activation quantization due to the actual speedup it provides with low-bit quantization kernels [1]. Quantizing LLMs faces challenges due to activation outlier features persisting across different tokens and layers [13, 64]. Some approaches [13, 74] retain a small portion of crucial outlier channels at high precision (e.g., INT8), which poses challenges to hardware compatibility and leads to additional memory footprint. Other methods [64, 59, 47] attempt to shift quantization difficulty from activation to weight channels. However, the learnable equivalent transformation in OmniQuant [47] and the affine transform matrix in AffineQuant [39] exhibit instability as discussed in Section 2. The channel disassembly and assembly in QLLM [34], coupled with LoRA-tuning, incur significant time costs. Notably, these methods demonstrate poor performance under W4A4 quantization. We attribute this degradation to the ineffective handling of outlier features, especially massive outliers. Hence, we propose DuQuant to effectively eliminate outlier features through rotation matrices and channel permutation, achieving state-of-the-art performance. In contrast with QuaRot [2] also utilizing Hadamard matrices to enhance weight-activation quantization, our approach uniquely incorporates knowledge about the actual outlier channels. Furthermore, unlike QuaRot, which relies on GPTQ [18] for weight quantization, our permutation transformation has been proven helpful and efficient, facilitating a faster quantization process. The more detailed analysis and comparison with QuaRot are left in Appendix F. In addition, unlike RPTQ [69] and SKVQ [17], which use channel reordering to cluster similar activations, our method employs Permutation transformations with a fundamentally different goal: to evenly distribute outliers across blocks. This balanced distribution is crucial for enabling effective secondary rotations, ultimately leading to smoother activations that facilitate easier quantization.

# B Proofs

**Theorem 1** (Rotation). *For the activation input* $\mathbf{X} \in \mathbb{R}^{T \times C_{in}}$, $\hat{\mathbf{R}} \in \mathbb{R}^{2^n \times 2^n}$ *is a diagonal block matrix constructed as per Eqn. (3). For a specific block* $b_i$, *let* $O_j(\cdot)$ *represent the maximum outlier of the* $j$-*th dimension* $d_j$ *within the input. Then, we can deduce that,*

$$\max_{1 \leq j \leq 2^n} O_j(\mathbf{X}_{b_i} \hat{\mathbf{R}}_{b_i}) \leq \max_{1 \leq j \leq 2^n} O_j(\mathbf{X}_{b_i}). \tag{8}$$

*Proof.* In the case of a specific block $b_i$, the potential maximum value $O_j(\mathbf{X}_{b_i} \hat{\mathbf{R}}_{b_i})$, where $\mathbf{X}_{b_i} \in \mathbb{R}^{T \times 2^n}$, can be achieved by ensuring that the different $j$-th column outliers $O_j(\mathbf{X}_{b_i})$ are located in the same $t$-th row of the activation input $\mathbf{X}_{b_i}$. This can be formally defined as follows,

$$\max_{1 \leq j \leq 2^n} O_j(\mathbf{X}_{b_i} \hat{\mathbf{R}}_{b_i}) = O_1(\mathbf{X}_{b_i} \mathbf{E}_{d^{(1)}}) \cdot \frac{1}{\sqrt{M}} + O_2(\mathbf{X}_{b_i} \mathbf{E}_{d^{(1)}}) \cdot \delta_1 + ... + O_M(\mathbf{X}_{b_i} \mathbf{E}_{d^{(1)}})\delta_M,$$

where $M = 2^n$ and $\|\frac{1}{2^n} + \delta_2^2 + ... + \delta_M^2\| = 1$,   ($\hat{\mathbf{R}}_{b_i}$ is orthogonal)

Without loss of generality, let's assume that $\delta_i \geq 0$ and define $m := \arg\max_i \delta_i^2$. Then we can derive that $\delta_m \geq \frac{1}{\sqrt{2^n}}$. Consequently, we can obtain the following inequality,

$$\begin{aligned}
\max_{1 \leq j \leq 2^n} O_j(\mathbf{X}_{b_i} \hat{\mathbf{R}}_{b_i}) &= O_1(\mathbf{X}_{b_i} \mathbf{E}_{d^{(1)}}) \cdot \frac{1}{\sqrt{2^n}} + O_2(\mathbf{X}_{b_i} \mathbf{E}_{d^{(1)}}) \cdot \delta_1 + ... + O_M(\mathbf{X}_{b_i} \mathbf{E}_{d^{(1)}})\delta_M \\
&\leq O_1(\mathbf{X}_{b_i} \mathbf{E}_{d^{(1)}}) \cdot \frac{1}{\sqrt{2^n}} + (O_2(\mathbf{X}_{b_i} \mathbf{E}_{d^{(1)}}) + ... + O_M(\mathbf{X}_{b_i} \mathbf{E}_{d^{(1)}}))\delta_m \\
&\overset{(1)}{\leq} O_1(\mathbf{X}_{b_i} \mathbf{E}_{d^{(1)}}) \cdot \frac{1}{\sqrt{2^n}} + O_1(\mathbf{X}_{b_i} \mathbf{E}_{d^{(1)}}) \cdot \frac{\sqrt{2^n} - 1}{\sqrt{2^n}} = O_1(\mathbf{X}_{b_i} \mathbf{E}_{d^{(1)}}) \\
&= \max_{1 \leq j \leq 2^n} O_j(\mathbf{X}_{b_i}).
\end{aligned} \tag{9}$$

The inequality (1) holds because the switch matrix $\mathbf{E}_{d^{(1)}}$ has swap the largest outliers $O_1(\mathbf{X}_{b_i})$ in the first column, i.e., $O_1(\mathbf{X}_{b_i} \mathbf{E}_{d^{(1)}}) > \max_{2 \leq m \leq M} O_m(\mathbf{X}_{b_i} \mathbf{E}_{d^{(1)}})$.

$\square$

**Theorem 2** (Zigzag Permutation). *For the activation input* $\mathbf{X} \in \mathbb{R}^{T \times C_{in}}$, *it can be divided into* $K$ *blocks, where* $K = C_{in}/2^n$. *Let* $O_j$ *denote the max outlier of the dimension* $d_j$ *in* $\mathbf{X}$, *the reordered outliers from large to small is expressed as* $O^{(1)}, O^{(2)}, ..., O^{(C_{in})}$. *Moreover, the* $M_{b_i}$ *represents the mean value of all* $O_j$ *in the* $i$-*th block,* $i = 1, 2, ..., K$. *Let* $\delta := \max\{|O^{(i+1)} - O^{(i)}|\}, i = 1, 2, ..., C_{in}-1$. *Then, following the zigzag permutation described in Section 3.2, the mean value* $M_{b_i}$ *within each* $i$-*th block consistently satisfies,*

$$M_{b_i} \leq O^{(1)} + \frac{(2^n K - 1)(2^{n-1} - 1)}{2^n}\delta, \qquad i = 1, 2, 3, ..., K. \tag{10}$$

*Proof.* According to the zigzag permutation described in Section 3.2, and considering the reordered outliers $O^{(1)}, O^{(2)}, ..., O^{(C_{in})}$, we can redistribute the channels they occupy (i.e., $O_c^{(i)}$)across different blocks. Specifically, for the $i$-th block, it contains the following channels,

$$\begin{aligned}
b_i &= \{O_c^{2mK+i}, O^{2(m+1)K-i+1} | m = 0, 1, ..., 2^{n-1} - 1\} \\
&= \{O_c^{(i)}, O_c^{(2K-i+1)}, ..., O_c^{2^n K + K}, O^{(2^n K + K + 1)}\}
\end{aligned}$$

Since $\delta = \max\{|O^{(i+1)} - O^{(i)}|\}, i = 1, 2, ..., C_{in}-1$, then we can get

$$\begin{aligned}
M_{b_1} &= \frac{1}{2^n}\{O^{(1)} + O^{(2k)} + ... + O^{2^n - 2k + 1} + O^{2^n K}\} \\
&\leq \frac{1}{2^n}\{O^{(1)} + (4K - 1)\delta + (8K - 1)\delta + [(2^{n+1} - 4)K - 1]\delta + (2^n K - 1)\delta\} \tag{11} \\
&\leq O^{(1)} + \frac{(2^n K - 1)(2^{n-1} - 1)}{2^n}\delta.
\end{aligned}$$

Similarly, we can deduce that all $M_{b_i}$ $(i = 1, 2, ..., K)$ share the same upper bound after applying our zigzag permutation.

$\square$

# C  Additional Implementation Details

In this work, all experiments are done on NVIDIA RTX 3090 GPUs for small-scale models and NVIDIA A100 GPUs for large-scale models. We set sequence length to 2048 for all evaluation tasks.

For calibration data, following [47, 39, 34], we randomly select 128 sampled sequences from the WikiText2 dataset, with the sequence length of 2048. For rotation and permutation transformations, the rotation block size $2^n$ is set to 128, and maximum greedy search steps $N$ equals 256. We adopt once permutation times for efficiency. We conduct detailed ablation studies in Appendix E.2, E.3, E.4.

Regarding quantization details, for multiplications between activations in MSA, such as Query and Key, attention outputs and Value, we apply a Hadamard rotation matrix for rapid and straightforward processing. A Hadamard matrix is an orthogonal and symmetric matrix filled with elements $\pm 1/\sqrt{2^n}$. For smooth parameter $\alpha$, we set it to 0.6 for DuQuant and 0.5 DuQuant+LWC. We clip the maximum activation values in all projection blocks, and the clipping ratio is set to 0.9. For DuQuant we also clip the maximum values in weight matrices, with a clipping ratio of 0.8. For LWC, we keep the same default epoch numbers of 20, batch size as 1, learning rate as 5e-3, and zero weight decay, as [47].

# D  More Empirical Results

Table D1: Zero-shot common-sense QA (↑) results of LLaMA2 models under 4-bit WA quantization.

| Model | Method | PIQA | ARC-E | ARC-C | BoolQ | HellaSwag | WinoGrande | Avg. |
|---|---|---|---|---|---|---|---|---|
| | FP16 | 76.88 | 53.54 | 40.53 | 71.13 | 72.96 | 67.25 | 63.72 |
| LLaMA2-7B W4A4 | SmoothQuant | 60.17 | 35.23 | 27.13 | 57.92 | 37.08 | 49.57 | 44.52 |
| | OS+ | 63.11 | 39.10 | 28.84 | - | 51.30 | 45.93 | 45.66 |
| | OmniQuant | 65.61 | 44.28 | 30.38 | 62.66 | 53.51 | 51.85 | 51.38 |
| | AffineQuant | 67.36 | 44.23 | 31.91 | 62.75 | 54.38 | 55.18 | 52.64 |
| | QLLM | 67.68 | 44.40 | 30.89 | - | 58.45 | 56.59 | 51.60 |
| | Atom | 69.75 | 47.35 | 34.22 | 62.42 | 63.21 | 56.51 | 55.58 |
| | **DuQuant** | 75.24 | **51.89** | 36.77 | 67.86 | 69.54 | 62.12 | 60.57 |
| | **DuQuant+LWC** | **75.68** | 50.00 | **37.46** | **69.24** | **69.74** | **63.93** | **61.01** |
| | FP16 | 79.05 | 57.91 | 44.20 | 69.02 | 76.60 | 69.69 | 66.08 |
| LLaMA2-13B W4A4 | SmoothQuant | 62.30 | 40.28 | 30.72 | 60.49 | 42.24 | 49.96 | 47.67 |
| | OS+ | 64.47 | 41.46 | 32.17 | - | 59.30 | 51.38 | 49.76 |
| | OmniQuant | 69.80 | 47.22 | 33.79 | 65.47 | 59.34 | 55.49 | 55.19 |
| | AffineQuant | 68.55 | 47.64 | 32.34 | 66.97 | 59.97 | 55.07 | 55.09 |
| | QLLM | 70.46 | 48.48 | 34.39 | - | 62.80 | 55.41 | 54.31 |
| | Atom | 71.16 | 50.89 | 37.88 | 63.91 | 67.51 | 58.40 | 58.29 |
| | **DuQuant** | **77.31** | 55.60 | 41.55 | **66.61** | **73.68** | **66.06** | **63.47** |
| | **DuQuant+LWC** | 77.26 | **56.23** | **42.15** | 65.78 | **73.68** | 65.43 | 63.42 |
| | FP16 | 81.01 | 59.68 | 47.95 | 75.87 | 80.87 | 76.95 | 70.39 |
| LLaMA2-70B W4A4 | SmoothQuant | 64.09 | 41.84 | 32.00 | 58.56 | 54.21 | 51.07 | 50.30 |
| | OS+ | 66.16 | 42.72 | 34.90 | - | 56.93 | 52.96 | 50.73 |
| | QLLM | 74.27 | 50.59 | 37.20 | - | 71.62 | 59.43 | 58.62 |
| | **DuQuant** | 79.27 | 58.16 | 46.07 | 70.46 | 79.21 | **74.19** | 67.89 |
| | **DuQuant+LWC** | **79.82** | **59.76** | **46.76** | **73.12** | **79.38** | 74.11 | **68.83** |

**Zero-shot QA Results for 4-bit LLaMA2 Models.**   Table D1 showcases the zero-shot common-sense QA results for INT4 quantized LLaMA2 models. Our DuQuant method excels across various model sizes and datasets, demonstrating state-of-the-art performance in commonsense reasoning tasks. For example, DuQuant outperforms Atom by 5.43% for the LLaMA2-7B model and by 5.18% for the LLaMA2-13B model. In contrast to Atom [74], which relies on GPTQ for weight quantization and maintains 128 channels at INT8, thereby increasing memory usage, our method offers a rapid and more efficient weight-activation quantization solution through Rotation and Permutation.

**MMLU Results for 4-bit Vicuna-v1.5-7B.**   Vicuna-v1.5 models [10], fine-tuned from LLaMA-2 models using high-quality user-shared conversations, are considered state-of-the-art chatbots. Table D2 displays the INT4 quantization results for Vicuna-v1.5-7B on the MMLU benchmarks. In comparison to SmoothQuant, OmniQuant, and Atom, our DuQuant method exhibits the smallest

performance decline and maintains competitive capacities in both zero-shot and five-shot settings. These results demonstrate the effectiveness of DuQuant in generalizing to instruction-tuned models.

Table D2: Zero-shot and five-shot results on the MMLU benchmark for quantized Vicuna-v1.5-7B.

| Model | Method | MMLU (0 shot) ↑ | | | | | MMLU (5 shot) ↑ | | | | |
|---|---|---|---|---|---|---|---|---|---|---|---|
| | | STEM | Hums | Social | Others | Avg. | STEM | Hums | Social | Others | Avg. |
| Vicuna-v1.5-7B W4A4 | FP16 | 38.70 | 45.42 | 56.13 | 56.01 | 49.07 | 39.56 | 45.76 | 58.14 | 57.43 | 50.22 |
| | SmoothQuant | 27.10 | 25.16 | 27.40 | 26.71 | 26.59 | 25.22 | 25.06 | 24.99 | 26.68 | 25.49 |
| | OmniQuant | 27.20 | 24.00 | 27.14 | 25.08 | 25.86 | 29.39 | 24.95 | 27.30 | 24.80 | 26.39 |
| | Atom | 30.28 | 34.73 | 38.97 | 40.56 | 36.14 | 31.97 | 35.37 | 40.46 | 40.81 | 37.15 |
| | **DuQuant** | **35.85** | **42.66** | **52.03** | **51.23** | **45.44** | **38.90** | **42.57** | 51.80 | **51.23** | **46.13** |
| | **DuQuant+LWC** | 35.18 | 41.91 | 51.28 | 50.52 | 44.72 | 37.34 | 42.21 | **53.07** | 51.76 | 46.10 |

**Long-context Evaluation Results.** LongBench [4] is proposed to access the long-context generation ability of LLMs, which covers several key long-text application scenarios. We evaluate the 4-bit quantized Vicuna models on five different tasks. Specifically, Qasper, MultiFieldQA, and NarrativeQA (F1 score) are Single-Document QA tasks; DuReader (Rouge-L score) and 2WikiMultihopQA (F1 score) are Multi-Document QA tasks; QMSum, GovReport (F1 score) and MultiNews (Rouge-L score) are Summarization tasks; TREC (Accuracy CLS), TriviaQA (F1 score), and SAMSum (Rouge-L score) are Few-shot Learning tasks; and RepoBench-P (similarity score) is Code Completion task. Table D3 and Table D4 show that our DuQuant outperforms other baselines by a clear margin, maintaining the ability for long context generation tasks compared with FP16 models.

Table D3: Long-context generation results on the LongBench benchmark for 4-bit Vicuna-v1.5-7B.

| Vicuna-v1.5-7B | RepoBench-P | MultiFieldQA-en | GovReport | MultiNews | DuReader | 2WikiMQA | TriviaQA |
|---|---|---|---|---|---|---|---|
| FP16 | 48.23 | 38.30 | 27.93 | 26.91 | 25.53 | 18.02 | 82.59 |
| SmoothQuant | 25.92 | 4.66 | 2.62 | 6.05 | 4.24 | 2.02 | 1.62 |
| OmniQuant | 14.97 | 2.30 | 2.51 | 2.64 | 1.87 | 0.48 | 0.81 |
| Atom | 29.34 | 31.15 | 23.60 | 24.60 | 19.41 | **17.10** | 67.20 |
| **DuQuant** | **47.66** | **35.62** | **25.66** | **25.85** | **23.15** | 15.09 | **78.91** |

| Vicuna-v1.5-7B | QMSum | MultiFieldQA-zh | NarrativeQA | Qasper | SAMSum | TREC | Avg |
|---|---|---|---|---|---|---|---|
| FP16 | 21.07 | 32.56 | 14.96 | 23.27 | 41.06 | 66.00 | 35.88 |
| SmoothQuant | 2.00 | 0.88 | 1.75 | 4.11 | 1.55 | 15.00 | 5.57 |
| OmniQuant | 3.93 | 1.40 | 1.10 | 1.62 | 0.61 | 1.00 | 2.71 |
| Atom | 20.24 | 21.55 | 11.57 | 17.97 | 37.94 | 58.00 | 29.21 |
| **DuQuant** | **21.15** | **29.56** | **11.31** | **19.98** | **42.24** | **64.00** | **33.86** |

Table D4: Long-context generation results on the LongBench benchmark for 4-bit Vicuna-v1.5-13B.

| Vicuna-v1.5-13B | RepoBench-P | MultiFieldQA-en | GovReport | MultiNews | DuReader | 2WikiMQA | TriviaQA |
|---|---|---|---|---|---|---|---|
| FP16 | 43.08 | 42.69 | 28.43 | 26.53 | 27.57 | 29.40 | 86.81 |
| SmoothQuant | 11.57 | 1.64 | 2.81 | 3.54 | 6.71 | 1.39 | 1.83 |
| OmniQuant | 8.46 | 4.32 | 0.74 | 2.83 | 13.83 | 0.75 | 1.13 |
| Atom | 37.31 | 37.31 | 19.34 | 23.39 | 21.79 | 15.16 | 80.75 |
| **DuQuant** | **38.09** | **44.12** | **26.97** | **26.59** | **26.02** | **22.07** | **83.04** |

| Vicuna-v1.5-13B | QMSum | MultiFieldQA-zh | NarrativeQA | Qasper | SAMSum | TREC | Avg |
|---|---|---|---|---|---|---|---|
| FP16 | 21.24 | 40.44 | 15.41 | 24.41 | 41.97 | 68.00 | 40.64 |
| SmoothQuant | 2.95 | 0.82 | 0.97 | 2.18 | 0.35 | 1.50 | 4.21 |
| OmniQuant | 1.78 | 1.06 | 0.62 | 0.68 | 0.45 | 9.00 | 4.58 |
| Atom | 20.23 | 28.02 | 8.81 | 17.67 | 38.72 | 59.00 | 33.58 |
| **DuQuant** | **20.72** | **30.85** | **13.36** | **18.93** | **42.67** | **66.50** | **38.13** |

**Comparison with FP16 models on MT-Bench.** We conducted additional comparisons using the MT-Bench between our INT4 quantized models and the FP16 models. As shown in Table D5, for both 7B and 13B models, our DuQuant performs comparably to FP16, which further underscores the effectiveness of dual transformations in maintaining high accuracy even with reduced precision.

Table D5: More results on MT-Bench.

| DuQuant v.s. FP16 | Former Win | Tie | Former Loss |
|---|---|---|---|
| Vicuna-v1.5-7B | 36 | 56 | 68 |
| Vicuna-v1.5-13B | 43 | 53 | 64 |

**Results for 4-bit Mistral-7B and Phi2-2.8B.** We have extended the application of DuQuant to include Mistral [27] and Phi2 [26] under 4-bit WA quantization. From Table D6, we can observe that

DuQuant largely surpasses other baselines, particularly with Mistral-7B. Regarding the Phi2-2.8B model, it often experiences instability in matrix multiplication between queries and values, leading to overflow issues and posing great challenges to quantization. However, while DuQuant may not perform as well as FP models, it still significantly outperforms other baselines. In addition, we have visualized the massive outliers in the down projection layer of the Mistral-7B model and the feature space after our dual transformations. These visualizations are shown in Figure I9. It can be observed that our DuQuant perfectly eliminates these outliers. These results underscore the effectiveness of our dual transformation approach in addressing massive outliers across various types of LLMs.

Table D6: Perplexity results of Mistral-7B and Phi2-2.8B under 4-bit weight-activation quantization.

| Model | Method | WikiText2 | C4 | Model | Method | WikiText2 | C4 |
|---|---|---|---|---|---|---|---|
| Mistral-7B W4A4 | FP16 | 5.25 | 7.75 | Phi2-2.8B W4A4 | FP16 | 9.71 | 12.76 |
| | RTN | 306.26 | 300.07 | | RTN | 230.59 | 253.79 |
| | SmoothQuant | 100.59 | 158.02 | | SmoothQuant | 63.84 | 83.24 |
| | OmniQuant | 5490.31 | 6094.82 | | OmniQuant | NaN | NaN |
| | Atom | 8.65 | 12.43 | | Atom | 35.72 | 41.26 |
| | **DuQuant** | **5.86** | **8.48** | | **DuQuant** | **20.65** | **22.49** |

**Results for 4-bit LLaMA3-70B.** As LLaMA3 models have proven to be sensitive to quantization, we apply our DuQuant to the LLaMA3-70B and present the results in Table D7. Due to time constraints, we do not add learnable weight clipping. The results demonstrate that our DuQuant-quantized models outperform SmoothQuant by 12.9% on Commonsense QA tasks and significantly reduce perplexity across the WikiText2, C4, and PTB datasets. These improvements underscore the robustness of our DuQuant method when applied to the LLaMA3-70B model.

Table D7: Perplexity and QA results of LLaMA3-70B under 4-bit weight-activation quantization.

| #Bits | Method | WikiText2↓ | C4↓ | PTB↓ | PIQA | ARC-E | ARC-C | BoolQ | HellaSwag | WinoGrande | Avg.↑ |
|---|---|---|---|---|---|---|---|---|---|---|---|
| FP16 | - | 2.9 | 6.9 | 8.2 | 82.4 | 86.9 | 60.3 | 85.2 | 84.9 | 80.6 | 80.1 |
| LLaMA3-70B W4A4 | SmoothQuant | 9.6 | 16.9 | 17.7 | 76.9 | 75.8 | 43.5 | 64.4 | 62.9 | 58.9 | 63.7 |
| | **DuQuant** | **4.9** | **8.3** | **8.7** | **81.1** | **80.8** | **57.3** | **81.3** | **82.1** | **77.0** | **76.6** |

**W6A6 Quantization Results.** To thoroughly evaluate the effectiveness of our DuQuant models, we conduct comprehensive assessments under the W6A6 quantization setting. The perplexity results for language generation tasks are displayed in Table D8, while the zero-shot accuracy for Commonsense QA tasks is detailed in Tables D9 and D10. Our findings reveal that DuQuant not only surpasses other baselines but also achieves nearly lossless performance with FP16 models in these tasks. Interestingly, in several instances, DuQuant slightly outperforms DuQuant+LWC . This suggests that the Rotation and Permutation transformations alone are sufficient to create highly competitive quantized models under W6A6 settings, without the need for additional enhancements such as the learnable weight clipping (LWC) technique. These outcomes highlight the exceptional versatility and robustness of DuQuant across various quantization scenarios, confirming its potential as a leading solution in post-training quantization for large language models.

Table D8: Preplexity (↓) results on the WikiText2 and C4 datasets under 6-bit WA quantization.

| Dataset | #Bit | Method | 1-7B | 1-13B | 1-30B | 1-65B | 2-7B | 2-13B | 2-70B |
|---|---|---|---|---|---|---|---|---|---|
| WikiText2 | FP16 | - | 5.68 | 5.09 | 4.10 | 3.53 | 5.47 | 4.88 | 3.31 |
| | W6A6 | SmoothQuant | 6.03 | 5.42 | 4.55 | 3.88 | 6.20 | 5.18 | 3.69 |
| | | OmniQuant | 5.96 | 5.28 | 4.38 | 3.75 | 5.87 | 5.14 | 3.71 |
| | | QLLM | 5.89 | 5.28 | 4.30 | 3.73 | 5.91 | 5.08 | 3.55 |
| | | **DuQuant** | **5.73** | **5.13** | **4.14** | **3.57** | **5.53** | **4.92** | **3.35** |
| | | **DuQuant+LWC** | 5.74 | **5.13** | 4.15 | 3.60 | **5.53** | **4.92** | **3.35** |
| C4 | FP16 | - | 7.08 | 6.61 | 5.98 | 5.62 | 6.97 | 6.46 | 5.52 |
| | W6A6 | SmoothQuant | 7.47 | 6.97 | 6.34 | 5.99 | 7.76 | 6.76 | 5.88 |
| | | OmniQuant | 7.43 | 6.84 | 6.22 | 5.82 | 7.48 | 6.74 | 5.91 |
| | | QLLM | 7.34 | 6.82 | 6.17 | 5.80 | 7.31 | 6.71 | 5.76 |
| | | **DuQuant** | **7.12** | **6.64** | **6.00** | **5.64** | **7.03** | **6.50** | **5.54** |
| | | **DuQuant+LWC** | 7.13 | **6.64** | 6.01 | **5.64** | **7.03** | **6.50** | **5.54** |

Table D9: Zero-shot common-sense QA (↑) results of LLaMA1 models under 6-bit WA quantization.

| Model | Method | PIQA | ARC-E | ARC-C | BoolQ | HellaSwag | WinoGrande | Avg |
|-------|--------|------|-------|-------|-------|-----------|------------|-----|
| LLaMA1-7B W6A6 | FP16 | 77.47 | 52.48 | 41.46 | 73.08 | 73.00 | 67.07 | 64.09 |
| | SmoothQuant | 76.75 | 51.64 | 39.88 | 71.75 | 71.67 | 65.03 | 62.81 |
| | OS+ | 76.82 | 51.35 | 41.13 | 72.08 | 71.42 | 65.98 | 61.13 |
| | OmniQuant | 77.09 | 51.89 | 40.87 | 72.53 | 71.61 | 65.03 | 63.17 |
| | AffineQuant | 76.60 | 52.29 | 40.63 | 72.65 | 71.29 | 63.85 | 62.89 |
| | QLLM | 77.26 | 52.02 | 41.04 | - | 71.40 | 65.19 | 61.38 |
| | **DuQuant** | **77.53** | 51.47 | **41.13** | 72.78 | **72.76** | 66.69 | 63.73 |
| | **DuQuant+LWC** | 77.42 | **52.65** | 40.53 | 71.53 | 72.64 | **67.72** | **63.75** |
| LLaMA1-13B W6A6 | FP16 | 79.10 | 59.89 | 44.45 | 68.01 | 76.21 | 70.31 | 66.33 |
| | SmoothQuant | 77.91 | 56.60 | 42.40 | 64.95 | 75.36 | 69.36 | 64.43 |
| | OS+ | 78.29 | 56.90 | 43.09 | 66.98 | 75.09 | 69.22 | 64.92 |
| | OmniQuant | 78.40 | 57.28 | 42.91 | 67.00 | 75.82 | 68.27 | 64.95 |
| | QLLM | 77.91 | 57.70 | 42.92 | - | 75.02 | 69.14 | 64.54 |
| | **DuQuant** | 78.62 | **59.51** | **44.03** | **68.44** | 75.98 | **70.08** | **66.11** |
| | **DuQuant+LWC** | **79.16** | 59.39 | 43.69 | 68.10 | 75.81 | 69.06 | 65.87 |
| LLaMA1-30B W6A6 | FP16 | 80.08 | 58.92 | 45.47 | 68.44 | 79.21 | 72.53 | 67.44 |
| | SmoothQuant | 77.14 | 57.61 | 42.91 | 65.56 | 78.07 | 69.92 | 65.20 |
| | OS+ | 80.14 | 58.92 | 45.05 | 68.02 | 77.96 | 71.98 | 67.01 |
| | OmniQuant | 79.81 | 58.79 | 45.22 | 68.38 | 78.95 | 72.21 | 67.23 |
| | QLLM | 79.65 | 58.08 | 44.11 | - | 78.38 | **73.24** | 66.69 |
| | **DuQuant** | 79.43 | **59.34** | 44.54 | **70.15** | 78.89 | 72.77 | **67.52** |
| | **DuQuant+LWC** | **80.09** | 57.95 | **45.05** | 68.72 | **79.17** | 73.09 | 67.35 |
| LLaMA1-65B W6A6 | FP16 | 80.79 | 58.71 | 46.24 | 82.29 | 80.72 | 77.50 | 71.04 |
| | SmoothQuant | 80.25 | 57.92 | 45.50 | 80.22 | 80.18 | 74.76 | 69.80 |
| | OS+ | 79.67 | 55.68 | 45.22 | 80.02 | 78.03 | 73.95 | 68.76 |
| | OmniQuant | 81.01 | 58.12 | 46.33 | 80.64 | 79.91 | 75.69 | 70.28 |
| | QLLM | 80.14 | 57.79 | 45.05 | - | 79.74 | 74.59 | 67.46 |
| | **DuQuant** | **80.96** | **59.09** | **46.76** | **82.20** | **80.68** | **77.27** | **71.16** |
| | **DuQuant+LWC** | 80.63 | 58.00 | 46.50 | 82.08 | 80.49 | 76.87 | 70.76 |

Table D10: Zero-shot common-sense QA (↑) results of LLaMA2 models under 6-bit WA quantization.

| Model | Method | PIQA | ARC-E | ARC-C | BoolQ | HellaSwag | WinoGrande | Avg |
|-------|--------|------|-------|-------|-------|-----------|------------|-----|
| LLaMA2-7B W6A6 | FP16 | 76.88 | 53.54 | 40.53 | 71.13 | 72.96 | 67.25 | 63.72 |
| | SmoothQuant | 75.57 | 53.62 | 39.93 | 69.54 | 71.76 | 66.14 | 62.76 |
| | OS+ | 76.22 | 52.74 | 40.70 | - | 71.89 | 65.19 | 61.35 |
| | OmniQuant | 76.55 | 53.83 | 40.96 | 68.75 | 55.89 | 65.59 | 60.26 |
| | QLLM | 77.48 | 52.99 | 39.33 | - | 71.38 | 65.98 | 61.43 |
| | **DuQuant** | 76.99 | 52.99 | 40.87 | 70.40 | 72.49 | 67.32 | **63.51** |
| | **DuQuant+LWC** | 76.88 | 52.31 | 40.44 | 69.72 | 72.60 | 66.93 | 63.15 |
| LLaMA2-13B W6A6 | FP16 | 79.05 | 57.91 | 44.20 | 69.02 | 76.60 | 69.69 | 66.08 |
| | SmoothQuant | 78.29 | 57.41 | 43.86 | 69.50 | 75.02 | 66.93 | 65.17 |
| | OS+ | 78.29 | 59.13 | 43.34 | - | 75.37 | 67.56 | 64.74 |
| | OmniQuant | 78.24 | 57.58 | 43.86 | 71.10 | 75.52 | 68.35 | 65.78 |
| | AffineQuant | 78.35 | 57.58 | 43.34 | 66.73 | 74.71 | 68.59 | 64.88 |
| | QLLM | 78.78 | 58.29 | 43.77 | - | 75.10 | 68.43 | 64.87 |
| | **DuQuant** | 78.62 | 56.94 | 43.43 | 68.35 | 76.19 | 69.22 | 65.46 |
| | **DuQuant+LWC** | 78.94 | 57.95 | 44.11 | 68.81 | 76.17 | 68.98 | **65.83** |
| LLaMA2-70B W6A6 | FP16 | 81.01 | 59.68 | 47.95 | 75.87 | 80.87 | 76.95 | 70.39 |
| | SmoothQuant | 79.87 | 57.32 | 45.65 | 77.13 | 79.01 | 74.03 | 68.84 |
| | OS+ | 79.33 | 59.09 | 47.18 | - | 79.46 | 75.06 | 68.02 |
| | OmniQuant | 80.20 | 60.27 | 46.84 | - | 80.55 | 76.01 | 68.77 |
| | QLLM | 80.63 | 59.01 | 45.99 | - | 79.64 | 75.37 | 68.13 |
| | **DuQuant** | 80.96 | 59.39 | 47.27 | 77.34 | 80.70 | 76.40 | 70.34 |
| | **DuQuant+LWC** | 81.18 | 59.26 | 47.78 | 77.86 | 80.68 | 76.95 | **70.62** |

# E  More Ablation Studies

## E.1  Time Speedup and Memory Saving

The current generation of LLMs usually splits into pre-filling and decoding phases and deploys on two separate machines [43]. Here, we present more speedup and memory-saving results for these two phases achieved with the LLaMA2-7B model on a single NVIDIA RTX 3090 GPU. We set the input sequence length to 2048 and the decoding steps to 256. End-to-end results for time speedup and memory savings during the pre-filling stage are shown in Table E11 and E12. We can observe that DuQuant achieves a maximum speedup of $2.01\times$ during the pre-filling phase, with speedup increasing as the batch size grows. From Table E12, DuQuant demonstrates significant memory savings, effectively reducing memory usage by up to $3.20\times$ through quantization. For the decoding phase, we enlarge the batch size to 64 and measure speedup along with memory usage for one LLaMA2-7B layer, constrained by the 24 GB memory of the GPU. As shown in Table E13, DuQuant maintains speedup and memory usage comparable to QuaRot. These results underscore the efficiency of DuQuant in optimizing resource utilization, highlighting its potential to enhance performance and reduce costs in deploying large language models, particularly in resource-constrained environments.

Table E11: End-to-end pre-filling speedup on LLaMA2-7B model.

| Batch Size | FP16 Time | DuQuant Time | Speedup |
|:---:|:---:|:---:|:---:|
| 1 | 568ms | 294ms | $1.93\times$ |
| 2 | 1003ms | 509ms | $1.97\times$ |
| 3 | 1449ms | 720ms | $2.01\times$ |

Table E12: Peak memory usage during pre-filling phase of LLaMA2-7B model.

| Batch Size | FP16 Mem. | DuQuant Mem. | Saving Factor |
|:---:|:---:|:---:|:---:|
| 1 | 15.28GB | 4.79GB | $3.20\times$ |
| 2 | 17.94GB | 5.94GB | $3.02\times$ |
| 3 | 20.56GB | 7.10GB | $2.90\times$ |

Table E13: Decoding phase results of one LLaMA2-7B layer with a batch size of 64.

| Method | Time (ms) | Saving Factor | Memory (GB) | Saving Factor |
|:---|:---:|:---:|:---:|:---:|
| FP16 | 659 | - | 3.550 | - |
| SmoothQuant | 437 | 1.508x | 1.669 | 2.127x |
| QLLM | OOM | - | OOM | - |
| QuaRot | 457 | 1.442x | 1.678 | 2.116x |
| DuQuant | 499 | 1.321x | 1.677 | 2.117x |

## E.2  Effects of Rotation Matrix

**Ablation of Rotation Block Size.**   To further explore the impact of rotation block size, we apply varying block sizes in the rotation matrices to both LLaMA2-7B and LLaMA2-13B models and evaluate the perplexity of the quantized models. The results, presented in Table E14, indicate that increasing block sizes generally improves model performance. This improvement occurs because larger block sizes allow outliers to be distributed across more channels, evening out values throughout the activation/weight matrix thereby enhancing quantization accuracy and performance. Additionally, quantization runtime decreases with larger block sizes, likely due to more efficient transformations during the reshaping of original activation/weight matrices. Consequently, we adopt 128 as our rotation block size for all experiments for efficiency and effectiveness.

**Ablation of Rotation Times.**   Identifying the optimal rotation matrix $\mathbf{R}$ is a complex challenge, so we employ a greedy search algorithm to approximate the matrix as $\hat{\mathbf{R}}$. We conduct an ablation study on the number of greedy steps $N$ and summarize the results in Table E15. Initially, as $N$ increases, the model performance improves, reflecting our ability to determine $\hat{\mathbf{R}}$ more effectively. However, when $N$ reaches 1024, the model begins to overfit. Consequently, we have chosen $N = 256$ for all our experiments, as it offers the optimal balance between model performance and time usage.

Table E14: Impact of rotation block size.

| Block Size | LLaMA2-7B | | | LLaMA2-13B | | |
|---|---|---|---|---|---|---|
| | WikiText2 ↓ | C4 ↓ | Time/s | WikiText2 ↓ | C4 ↓ | Time/s |
| 4 | 18.69 | 26.48 | 64.4 | 8.81 | 13.03 | 97.7 |
| 8 | 10.77 | 15.04 | 53.8 | 7.02 | 9.68 | 80.8 |
| 16 | 8.69 | 11.46 | 48.2 | 6.12 | 8.12 | 75.2 |
| 32 | 6.96 | 8.85 | 48.3 | 5.61 | 7.35 | 76.2 |
| 64 | 6.38 | 8.07 | 50.1 | 5.45 | 7.13 | 74.0 |
| 128 | 6.28 | 7.90 | 48.6 | 5.42 | 7.05 | 74.0 |

Table E15: Impact of rotation times.

| Rotation Times | LLaMA2-7B | | | LLaMA2-13B | | |
|---|---|---|---|---|---|---|
| | WikiText2 ↓ | C4 ↓ | Time/s | WikiText2 ↓ | C4 ↓ | Time/s |
| 1 | 6.60 | 8.41 | 22.9 | 5.48 | 7.12 | 37.7 |
| 4 | 6.34 | 8.04 | 22.6 | 5.41 | 7.06 | 38.7 |
| 16 | 6.32 | 7.98 | 28.8 | 5.43 | 7.05 | 41.8 |
| 64 | 6.34 | 7.98 | 29.0 | 5.43 | 7.06 | 47.0 |
| 256 | 6.28 | 7.90 | 48.6 | 5.42 | 7.05 | 74.0 |
| 1024 | 6.31 | 8.01 | 129.7 | 5.46 | 7.12 | 179.8 |

## E.3 Effects of Permutation Algorithm.

As discussed in Section 3.2, rotation transformations within each block are limited and unable to re-distribute outliers across different blocks. To address this, we introduce a permutation transformation aimed at balancing outliers more comprehensively. Our primary goal is to minimize the variance among different blocks, as outlined in Eqn. (4). We explore several optimization algorithms, with the results detailed in Table E16. Note that the variance values are measured on activation values of the query project in the first layer of each model, and the time in the table represents the runtime of calibration. The Zigzag permutation notably reduces the variance to 3.0e-4, achieving this with minimal time expenditure and yielding competitive perplexity results. While Simulated Annealing slightly outperforms Zigzag in terms of perplexity for the LLaMA2-7B model, it was significantly more time-consuming, and the marginal gains did not justify the additional complexity. Therefore, we select Zigzag permutation as our preferred method, leading to smoother outlier distribution and more effective quantized models.

Table E16: Impact of channel permutation algorithm.

| Permutation Method | LLaMA2-7B | | | | LLaMA2-13B | | | |
|---|---|---|---|---|---|---|---|---|
| | WikiText2 ↓ | C4 ↓ | Variance | Time/s | WikiText2 ↓ | C4 ↓ | Variance | Time/s |
| w.o. Permutation | 7.92 | 10.64 | 3.9e-2 | 27.5 | 5.96 | 7.94 | 3.1e-2 | 44.7 |
| Random | 6.40 | 8.08 | 4.9e-3 | 89.5 | 5.43 | 7.07 | 3.9e-3 | 148.6 |
| Simulated Annealing | 6.26 | 7.89 | 1.7e-4 | 769.6 | 5.42 | 7.06 | 1.5e-4 | 1257.8 |
| Zigzag | 6.28 | 7.90 | 3.0e-4 | 48.6 | 5.42 | 7.05 | 2.5e-4 | 74.0 |

## E.4 Effects of Calibration Datasets

**Ablation of Different Calibration Datasets.**  We apply our DuQuant to quantize the LLaMA2-7B model using different calibration datasets, with results presented in Table E17. It can be observed that the selection of calibration datasets has a relatively minor impact on quantization performance. This is because our method uses the calibration data solely to identify outlier channels, rather than for gradient-based parameter learning as seen in methods like OmniQuant [47] and AffineQuant [39]. This ablation study underscores the robustness of our DuQuant method.

Table E17: Ablation of calibration datasets.

| LLaMA2-7B | | Eval. | |
| | | WikiText2 ↓ | C4 ↓ |
|---|---|---|---|
| Calib. | WikiText2 | 6.28 | 7.90 |
| | C4 | 6.25 | 7.87 |

**Calibration-free Quantization.** To further explore the robustness of DuQuant under varying calibration conditions, we generate random calibration data within the vocabulary range of the model, setting the sample count to 256. The results, shown in Table E18, indicate that even in calibration-free settings, our method continues to perform well, achieving results that are competitively close to those obtained with actual calibration data. This demonstrates that DuQuant could provide a viable solution in real-world scenarios where obtaining specific calibration data is challenging or impossible. In addition, our findings suggest that outliers are inherent to certain model layers, reflecting characteristics of the model weights or modules, especially in recent LLMs. This aligns with concurrent research: [66] identified consistent massive outliers at the FFN down projection layer in GLU-based LLMs, such as LLaMA, Mistral, Mixtral, SOLAR, and Gemma, while [42] reported that, although OPT models are sensitive to different calibration sets, newer models demonstrate robustness to outliers and maintain stable activations. These insights reinforce the idea that outliers are more linked to the internal structure of model weights and modules than to calibration data. DuQuant's ability to deliver high performance without relying on traditional calibration data creates opportunities for deploying quantized models in environments with stringent privacy requirements or limited data availability. This highlights a promising direction for future research, focusing on improving model adaptability and deployment flexibility.

Table E18: Calibration-free quantization, where we generate random data within vocabulary range.

| LLaMA2-7B | | Eval. | |
| | | WikiText2 ↓ | C4 ↓ |
|---|---|---|---|
| Calib. | Randomly Generated | 6.25 | 7.86 |
| | WikiText2 | 6.25 | 7.87 |

| LLaMA2-13B | | Eval. | |
| | | WikiText2 ↓ | C4 ↓ |
|---|---|---|---|
| Calib. | Randomly Generated | 5.45 | 7.05 |
| | WikiText2 | 5.44 | 7.05 |

**Ablation of Different Numbers of Calibration Samples.** We utilize our DuQuant to quantize the LLaMA2-7B model using varying numbers of calibration samples from the WikiText2 dataset, with results detailed in Table E19. Interestingly, the quantization performance shows a low correlation with the number of samples, demonstrating the robustness of DuQuant. This stability arises because we utilize the mean activation values from these samples to construct our rotation matrices. Since we average the activations, the influence of any single, potentially non-representative sample is minimized, ensuring consistent performance. Notably, as we use mean values, the time cost of our quantization process remains constant regardless of the number of samples, enhancing the efficiency.

Table E19: Ablation of different numbers in the calibration dataset.

| # of Samples | WikiText2 ↓ | C4 ↓ |
|---|---|---|
| 16 | 6.29 | 7.88 |
| 32 | 6.31 | 7.99 |
| 64 | 6.29 | 7.88 |
| 128 | 6.28 | 7.90 |
| 256 | 6.23 | 7.88 |

# F Detailed Comparison with QuaRot

Table F20: Comparison of quantization settings between QuaRot and DuQuant.

| Setting | Weight | Activation | Query |
|---------|--------|------------|-------|
| QuaRot | per-channel symmetric | per-token symmetric | FP16 |
| DuQuant | per-channel asymmetric | per-token asymmetric | per-token asymmetric |

Table F21: Evaluation results between QuaRot and DuQuant under DuQuant settings.

| Model | Method | WikiText2↓ | c4↓ | PIQA | ARC-E | ARC-C | BoolQ | HellaSwag | WinoGrande | Avg↑ |
|-------|--------|-----------|-----|------|-------|-------|-------|-----------|------------|------|
| | FP16 | 5.68 | 7.08 | 77.47 | 52.48 | 41.46 | 73.08 | 73.00 | 67.07 | 64.09 |
| LLaMA1-7B | QuaRot-RTN | 7.08 | 8.73 | 74.59 | 48.57 | 36.01 | 68.99 | 65.69 | 58.56 | 46.03 |
| W4A4 | QuaRot-GPTQ | 6.44 | 7.87 | 76.17 | 49.96 | 38.23 | 70.80 | 69.29 | 63.06 | 61.25 |
| | **DuQuant** | 6.40 | 7.84 | **76.44** | **50.04** | **38.99** | **70.98** | 69.39 | **64.72** | **61.76** |
| | **DuQuant+LWC** | **6.18** | **7.73** | 76.22 | 50.04 | 38.31 | 70.09 | **69.82** | 62.59 | 61.18 |
| | FP16 | 5.47 | 6.97 | 76.88 | 53.54 | 40.53 | 71.13 | 72.96 | 67.25 | 63.72 |
| LLaMA2-7B | QuaRot-RTN | 9.66 | 11.98 | 69.48 | 46.25 | 32.76 | 64.80 | 60.75 | 56.67 | 44.04 |
| W4A4 | QuaRot-GPTQ | 6.39 | 8.15 | 75.15 | 49.15 | 36.68 | 67.89 | 68.87 | 61.33 | 59.85 |
| | **DuQuant** | 6.28 | 7.90 | 75.24 | **51.89** | 36.77 | 67.86 | 69.54 | 62.12 | 60.57 |
| | **DuQuant+LWC** | **6.08** | **7.79** | **75.68** | 50.00 | **37.46** | **69.24** | **69.74** | **63.93** | **61.01** |
| | FP16 | 6.14 | 8.88 | 80.85 | 77.78 | 53.41 | 81.28 | 79.16 | 72.84 | 74.22 |
| LLaMA3-8B | QuaRot-RTN | 13.89 | 17.59 | 69.64 | 57.58 | 34.56 | 66.76 | 63.46 | 62.75 | 59.13 |
| W4A4 | QuaRot-GPTQ | 8.69 | 12.40 | 74.54 | 67.38 | 40.61 | 70.43 | 70.47 | 65.11 | 64.76 |
| | **DuQuant** | 8.53 | 12.01 | **76.93** | **70.88** | **45.05** | **74.59** | 73.17 | 66.14 | **67.79** |
| | **DuQuant+LWC** | **8.06** | **11.29** | 76.22 | 70.41 | 43.69 | 74.34 | **73.87** | **67.80** | 67.72 |

Table F22: Evaluation results between QuaRot and DuQuant under QuaRot settings.

| Model | Method | WikiText2↓ | C4↓ | PIQA | WinoGrande | HellaSwag | ARC-E | ARC-C | LAMBADA | Avg↑ |
|-------|--------|-----------|-----|------|------------|-----------|-------|-------|---------|------|
| | FP16 | 5.47 | 6.97 | 79.11 | 69.06 | 75.99 | 74.58 | 46.25 | 73.90 | 69.82 |
| LLaMA2-7B | QuaRot-RTN | 8.37 | - | 72.09 | 60.69 | 65.4 | 58.88 | 35.24 | 57.27 | 58.26 |
| W4A4 | QuaRot-GPTQ | 6.1 | - | 76.77 | 63.77 | 72.16 | 69.87 | 40.87 | **70.39** | 65.64 |
| QuaRot Setting | **DuQuant** | 6.23 | 7.91 | 76.28 | 66.93 | 72.96 | 69.99 | 40.53 | 69.61 | 66.05 |
| | **DuQuant+LWC** | **6.01** | **7.67** | **77.64** | **67.8** | **72.97** | **70.37** | **41.81** | 69.53 | **66.69** |
| | FP16 | 4.88 | 6.46 | 80.47 | 72.22 | 79.39 | 77.48 | 49.23 | 76.75 | 72.59 |
| LLaMA2-13B | QuaRot-RTN | 6.09 | - | 77.37 | 67.32 | 73.11 | 70.83 | 43.69 | 70.66 | 67.16 |
| W4A4 | QuaRot-GPTQ | 5.4 | - | 78.89 | 70.24 | 76.37 | 72.98 | 46.59 | 73.67 | 69.79 |
| QuaRot Setting | **DuQuant** | 5.39 | 7.05 | 78.51 | **70.88** | 76.80 | **74.62** | **48.21** | 73.92 | **70.49** |
| | **DuQuant+LWC** | **5.27** | **6.93** | **78.73** | **70.88** | **77.20** | 74.07 | 47.27 | **73.96** | 70.35 |

Table F23: Matrices comparison between DuQuant and QuaRot under W4A4 quantization.

| Model | | LLaMA2-7B | | LLaMA2-13B | |
|-------|---|-----------|---|------------|---|
| Dataset | | WikiText2↓ | C4↓ | WikiText2↓ | C4↓ |
| QuaRot | | 9.66 | 11.98 | 6.73 | 8.69 |
| **DuQuant** | | **7.92** | **10.64** | **5.96** | **7.94** |

Table F24: Quantization runtime comparison on a single NVIDIA A100 80G GPU.

| Model | LLaMA2-7B | LLaMA2-13B | LLaMA2-70B |
|-------|-----------|------------|------------|
| QuaRot | 20min | 36min | 5.1h |
| **DuQuant** | 50s | 71s | 270s |

In this section, we present a detailed comparison between our DuQuant and QuaRot [2]. QuaRot employs Hadamard matrices to mitigate outliers in activations and utilizes the GPTQ algorithm for weight quantization to achieve competitive performance. However, our DuQuant method demonstrates several distinct advantages:

- **Effective Use of Prior Knowledge:** DuQuant leverages prior knowledge to accurately target and eliminate outliers through multiple rotations, achieving a smoother activation distribution compared to the Hadamard transformation, as demonstrated in Figure 10.

- **Efficient Channel Permutation:** Our channel permutation not only further smooths outlier features but also benefits from rapid implementation, enhancing overall performance.

- **Simultaneous Weight Matrix Smoothing:** Unlike QuaRot, DuQuant directly and efficiently smooths the weight matrix, avoiding the time-consuming GPTQ algorithm and accelerating the quantization process, as demonstrated high quantization efficiency in Table F24.

Experimental results underscore the superiority of DuQuant over QuaRot. We first summarize the experimental setting differences between the original paper of QuaRot with ours in Table F20*. For a fair comparison, we reproduce the QuaRot under 4-bit per-channel weight and per-token activation asymmetric quantization. Table F21 displays the perplexity (PPL) and zero-shot accuracy for models LLaMA1-7B, LLaMA2-7B, and LLaMA3-8B. Our DuQuant method consistently outperforms QuaRot-RTN across all benchmarks, showcasing our advanced weight matrix management. Furthermore, compared to QuaRot-GPTQ, DuQuant and DuQuant+LWC achieve better average accuracy across six QA tasks and demonstrate superior performance on the WikiText and C4 datasets, particularly for LLaMA3-8B. Moreover, we further provide the results of DuQuant under the setting utilized in the original paper of QuaRot in Table F22. DuQuant still surpasses QuaRot by a large margin.

Additionally, we assess the effectiveness of the rotation matrices utilized in DuQuant which incorporate prior knowledge against the Hadamard matrices used in QuaRot. We omit the permutation step in DuQuant and directly contrast it with QuaRot-RTN. Results in Table F23 show that our DuQuant without permutation outperforms QuaRot by a clear margin, which confirms that our rotation transformation is more effective than Hadamard by leveraging prior knowledge. It is worth noting that because a Hadamard matrix is orthogonal and symmetric, it multiplies by itself to yield the identity matrix. In other words, the Hadamard matrix is not suitable for greedy searches aimed at finding smaller outliers. These findings differentiate DuQuant from QuaRot and highlight the effectiveness of our approach in managing outliers for post-training quantization of large language models.

# G   Algorithm for Rotation Matrix

---

**Algorithm 1** Construction of the Rotation Matrix

---

**Input:** Pre-initialized rotation matrix $\tilde{\mathbf{R}}$, greedy search steps $N$, activation matrix $\mathbf{X}$ with shape of $[T, C_{\text{in}}]$
**Output:** Rotation matrix $\hat{\mathbf{R}}$
**function** get_rotation_matrix $(\mathbf{X}, \tilde{\mathbf{R}}, N)$

1: $T, C_{\text{in}} = \mathbf{X}.\text{shape}$
2: $\mathbf{R} = \text{eye}(C_{\text{in}})$
3: $a = \max_{i,j} |\mathbf{X}_{ij}|$
4: **for** $k$ in $1, ..., N$ **do**
5:      channel_max = $\mathbf{X}.\text{abs}().\text{max}(\text{dim} = 0).\text{values}$
6:      outlier_channel = $\arg\max(\text{channel\_max})$
7:      Obtain randomly initialized orthogonal matrix $\mathbf{Q}'$ with shape of $[C_{\text{in}} - 1, C_{\text{in}} - 1]$
8:      $\mathbf{Q}' = \text{concat}([\text{zeros}(C_{\text{in}} - 1, 1), \mathbf{Q}'], \text{dim}=1)$
9:      $\mathbf{Q} = \text{concat}([\text{zeros}(1, C_{\text{in}}), \mathbf{Q}'], \text{dim}=0)$
10:      $\mathbf{Q}[0, 0] = 1$
11:      $\mathbf{R}' = \text{matmul}(\tilde{\mathbf{R}}, \mathbf{Q})$
12:      $\mathbf{R}'[:, \text{outlier\_channel}], \mathbf{R}'[:, 0] = \mathbf{R}'[:, 0], \mathbf{R}'[:, \text{outlier\_channel}]$
13:      $\mathbf{R}'[\text{outlier\_channel}, :], \mathbf{R}'[0, :] = \mathbf{R}'[0, :], \mathbf{R}'[\text{outlier\_channel}]$
14:      $\mathbf{R} = \text{matmul}(\mathbf{R}, \mathbf{R}')$
15:      $\mathbf{X} = \text{matmul}(\mathbf{X}, \mathbf{R}')$
16:      **if** $\max_{i,j} |\mathbf{X}_{ij}| < a$ **then**
17:          $\hat{\mathbf{R}} = \mathbf{R}$
18:          $a = \max_{i,j} |\mathbf{X}_{ij}|$
19:      **end if**
20: **end for**
21: **return** $\hat{\mathbf{R}}$

---

*Following prior works [47, 34, 39], we describe operations on Query, Key, and Value states as per-tensor quantization for consistency. To be precise, these operations are effectively applied on a per-head basis.

## H  Limitations and Broader Impacts

**Limitations.**    The primary limitation of our method is the lack of a specialized strategy for calibration data selection. We adhere to established practices [47, 39, 34, 74, 2, 1] by randomly selecting 128 samples from the WikiText2 dataset to compute the mean embeddings that inform our rotation matrix and zigzag permutation order. We also explore the possibility of calibration-free quantization and show some promising results. However, further investigating more tailored choices for calibration data can potentially enhance the performance of our quantized models. We leave this for future study.

**Broader Impacts.**    Our work identifies the presence of massive outliers in down-projection layer of FFN modules, which significantly complicates low-bit weight-activation quantization. To address this challenge, we implement a combination of rotation and permutation matrices to effectively smooth both massive and uniform outliers, proving both fast and effective. Consequently, we establish a new state-of-the-art for INT4 weight-activation post-training quantization methods. Our approach aims to accelerate large language models and reduce memory usage during deployment, offering substantial benefits to the field of LLM research. These advancements could lead to more efficient and accessible LLM applications, facilitating broader usage and enabling more sustainable AI implementations.

## I  More Visualizations

We provide additional visualizations of normal and massive outliers in various models (LLaMA1, LLaMA2, Vicuna-v1.5, Mistral) from Figure I1 to Figure I9. In each figure except Mistral, the left side illustrates changes in normal outliers before and after applying our rotation and permutation transformations, while the right side shows the changes in massive outliers before and after transformations. It is evident that massive outliers consistently occur in the down-projection layer of the FFN module across all models, supporting our findings discussed in Section 2. Conversely, normal outliers appear in different modules within the transformation block. For instance, Figure I3 shows normal outliers at the up-projection layer of the FFN module in LLaMA1-13B. Significantly, both massive and normal outliers are reduced markedly after our rotation and permutation transformations, leading to easier quantization of activations. This underscores the effectiveness of our DuQuant in managing outlier features across diverse LLM models.

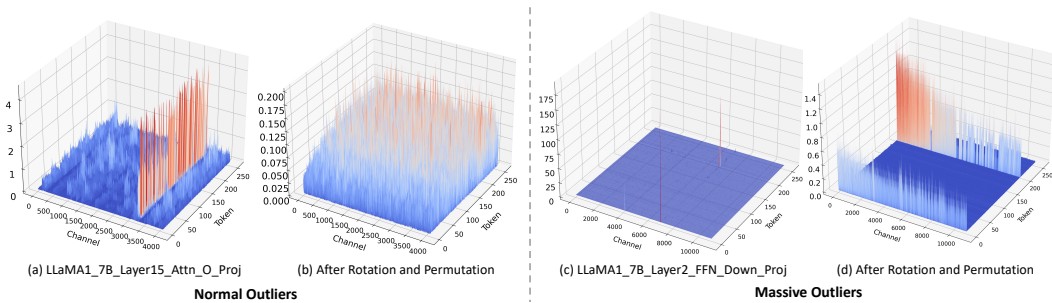

Figure I1: Activation change with the use of our DuQuant for LLaMA1-7B.

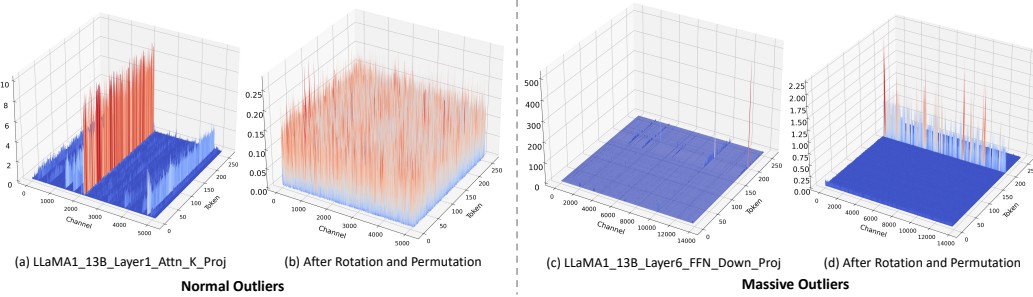

Figure I2: Activation change with the use of our DuQuant for LLaMA1-13B.

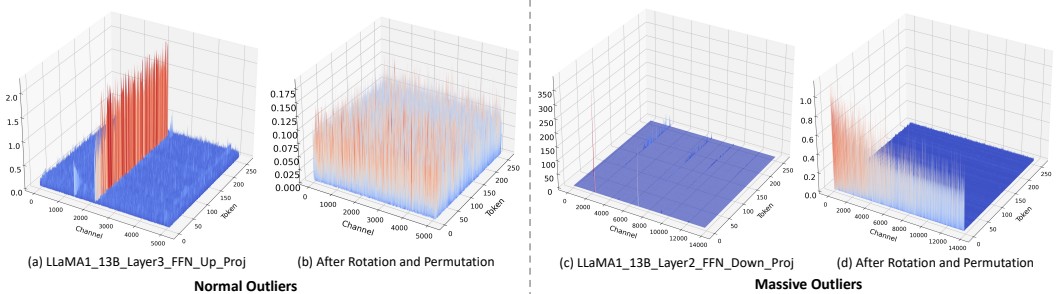

Figure I3: More examples of activation change with the use of our DuQuant for LLaMA1-13B.

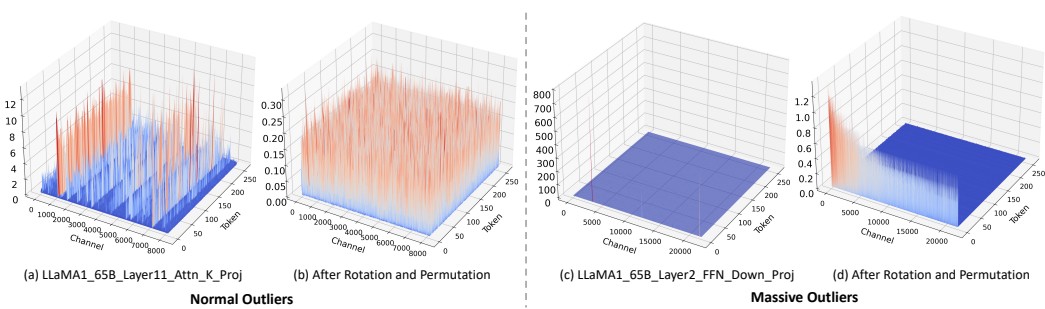

Figure I4: Activation change with the use of our DuQuant for LLaMA1-65B.

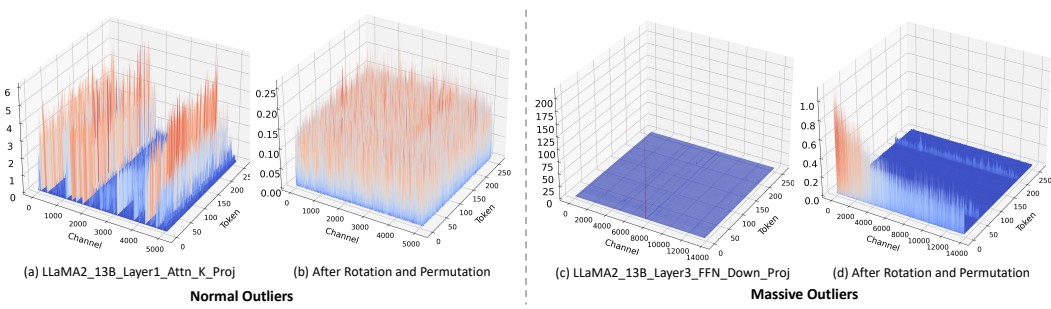

Figure I5: Activation change with the use of our DuQuant for LLaMA2-13B.

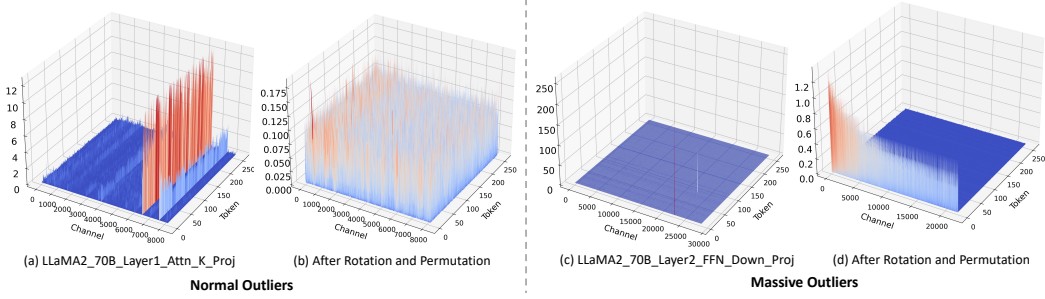

Figure I6: Activation change with the use of our DuQuant for LLaMA2-70B.

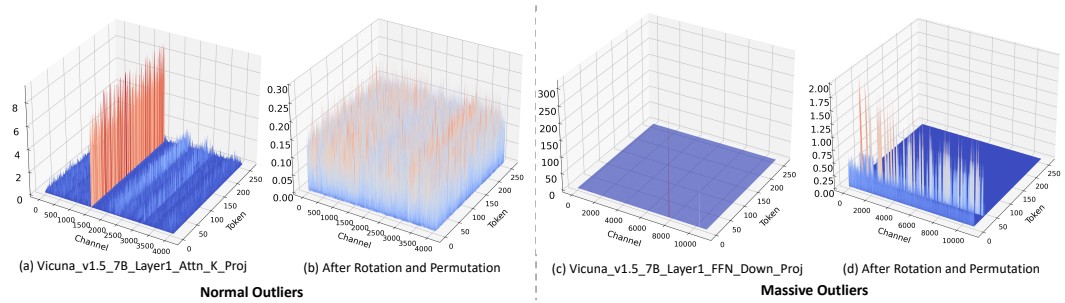

Figure I7: Activation change with the use of our DuQuant for Vicuna-v1.5-7B.

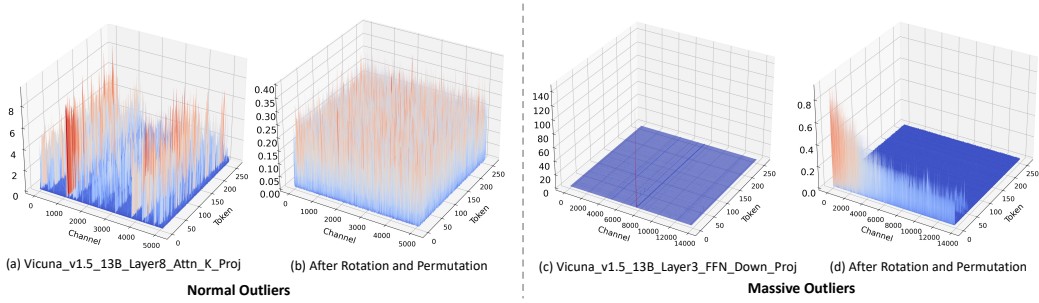

Figure I8: Activation change with the use of our DuQuant for Vicuna-v1.5-13B.

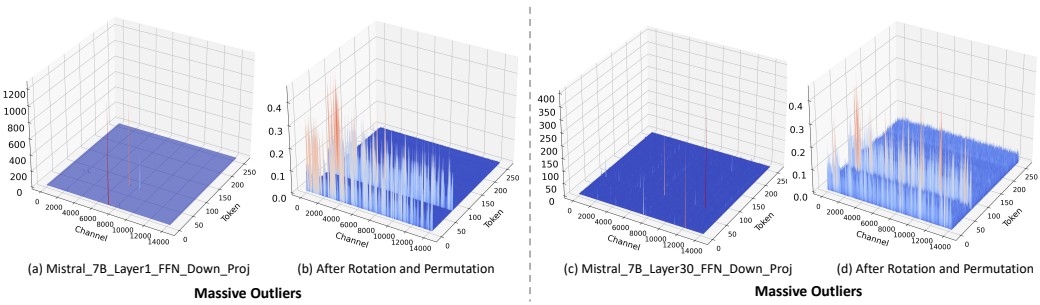

Figure I9: Massive activation change with the use of our DuQuant for Mistral7B.

