# OpenReview forum: "DuQuant: Distributing Outliers via Dual Transformation Makes Stronger Quantized LLMs"
_NeurIPS.cc/2024/Conference — NeurIPS 2024 oral_

### Official Review · Reviewer_Rbrg · 2024-06-15

**Soundness:** 3
**Presentation:** 4
**Contribution:** 3
**Rating:** 8
**Confidence:** 3

**Summary:**

The paper introduces a method called DuQuant, a new quantization method specialized for LLMs. The paper notes that "massive outliers" cause previous quantization approaches to be less effective or powerful, and then proposes a new quantization method which ameliorates the effect of such massive outliers. Theoretical analysis shows that the method has good properties, such as successfully spreading out the outlier mass directly before quantization, which facilitates better outcomes. Experiments show state-of-the-art performance across standard language modelling benchmarks on Llama and Llama-2 class models, preserving most of the base model's performance.

**Strengths:**

- The exposition/writing is very clear.
- All major steps of the method are motivated, and the theoretical calculations straightforwardly apply to the method's real-world implementation. The design choices are either made straightforwardly or chosen via experiments/ablations.
- Experimental results show that the overall model performance is better than other (previously state-of-the-art) quantization methods. The time and memory costs are also reduced, owing to the simple construction of the matrices in the construction (block diagonal, orthogonal, or permutation matrices).

**Weaknesses:**

Several potentially desirable properties of the model may not be completely covered by the analysis. For example:
- The memory consumption reduction seems good, but the presentation of this result is postponed to the appendix E. Could you find a way to put this in the main body (maybe moving back the presentation of one of the several pure performance metrics such as PPL, or one of the ablation studies), and compare to other state-of-the-art models? In my opinion, since people usually only do quantization for better memory consumption or speeding up inference, measuring the performance on both of these axes is crucial.
- The speedup is only measured for pre-filling. Is it possible to compute end-to-end speedup for open-ended generation? It would be great to see the impact of the method on some more realistic workloads (again, across the axes of performance, memory consumption, and speedup), to supplement the thorough analysis of each component in the paper.
To address both these issues, along with an empirical study, an analysis of the asymptotic runtime/memory complexity of the quantization procedure would be great if possible.

Also a slight nitpick: at the bottom of page 2, massive activations are not the same phenomenon as attention sinks, though they can be related.

**Questions:**

- Previous work on Massive Activations have shown their ubiquity in many types of LLMs, not just Llama, and even on certain types of vision models. Is it possible that this work can be done on other types of LLMs/transformers? Does the backbone model you pick significantly change the effectiveness of the method (e.g., Phi vs Llama vs Mistral)?

**Limitations:**

The authors are generally precise about the method's strengths and weaknesses, and include a limitations section.

---

> ### Author Rebuttal · Authors · 2024-08-07
>
> Thank you sincerely for your thoughtful and positive feedback on our work. We are particularly grateful for your recognition of the various aspects of our research. Below, we have provided a detailed explanation for your remaining concern as follows. Please do not hesitate to let us know if you have any further questions.
>
> **W1**: Comparison of memory reduction.
>
> > - Thanks for the suggestion. We will move the memory consumption reduction results to the main body for better visibility. Additionally, to provide a more comprehensive comparison, we have conducted evaluations of our DuQuant alongside SmoothQuant [1], QLLM [2], and QuaRot [3] using one RTX 3090 GPU during the prefilling stage. These comparisons will be detailed in the revised manuscript to highlight the relative efficiencies of each method.
> >
> >   |LLaMA2-7B, INT4, BS=1|Prefilling Memory (GB)|Saving Factor|
> >   |-| -| -|
> >   |FP16|15.282|-|
> >   |SmoothQuant |4.782| 3.196x|
> >   |QLLM|5.349|2.857x|
> >   |QuaRot|4.784|3.194x|
> >   |DuQuant|4.786|3.193x|
> >
> > - From the table, we can observe that 4-bit quantization methods can effectively reduce memory usage during the pre-filling stage.  DuQuant, SmoothQuant, and QuaRot achieve **significant reductions up to 3.2x**, while the QLLM performs much worse.
> >
> >  [1] Smoothquant: Accurate and efficient post-training quantization for large language models, ICML, 2023.
> >
> >  [2] QLLM: Accurate and Efficient Low-Bitwidth Quantization for Large Language Models, ICLR, 2024.
> >
> >  [3] Quarot: Outlier-free 4-bit inference in rotated llms, arXiv 2024.
>
> **W2**: Analysis of model performance, memory reduction, and inference speedup.
>
> > - Thank you for your valuable comment. LLM generation process includes a pre-filling stage, which is compute-bound, along with a decoding stage, which is memory-bound [1]. To comprehensively analyze these procedures, we will list a table in the revised manuscript comparing model performance with key features such as decoding memory usage and pre-filling time. For pre-filling, we measure time usage by sending one sentence with 2048 tokens and we decode 128 steps to compute peak memory usage. All the experiments are conducted on a single RTX 3090 GPU.
> >
> >   |INT4, BS=1|Time (ms)|Saving Factor|Memory (GB)|Saving Factor|WiKi↓|QA  avg.↑|
> >   |-|-|-|-|-|-|-|
> >   |FP16|568| -| 13.638|-| 5.47| 63.72 |
> >   |SmoothQuant | 248 | 2.290x| 3.890| 3.506x| 83.12|44.52|
> >   |QLLM|435|1.306x | 3.894| 3.502x| 9.09| 51.60|
> >   |QuaRot|284|2.000x | 3.891| 3.505x| 6.39| 61.25|
> >   |DuQuant|288|1.972x| 3.893| 3.503x| 6.28|61.76|
> >
> > - From the table, we can observe that our DuQuant effectively speeds up the pre-filling stage and largely reduces memory usage during decoding stage while providing **the most competitive results**.
> > - Due to time constraints and the significant workload involved, we couldn't test all methods on real-world generative tasks that require complex CUDA kernel optimizations. However, we are committed to continually optimizing DuQuant to improve its speed and efficiency in future updates.
> > - In addition, we present a simple **time complexity analysis** of our quantization process. We denote activation as $X\in\mathbf{R}^{N\times C}$, block size as $B$, and greedy search step size as $n$. The complexity of obtaining the rotation matrix of a linear projection is caused by (1) QR decomposition $O(nB^3)$, (2) Rotation matrices multiplication $O(nB^3)$, and (3) Multiplication between $X$ and rotation matrics $O(n\times NC/B \times B \times B)=O(nNCB)$. Thus, the total complexity is $O(nB^3+nNCB)$. For example, in LLaMa2-7B k_proj, we take $N=2048, C=4096$. We set $B=128,n=256$, then we can get the approximate computation complexity $nB^3+nNCB\approx 2.7\times 10^{11}$, which is less than the necessary WA multiplication complexity (approximately equal to $NC^2\approx 3.4\times 10^{11})$. This simple analysis demonstrates the efficiency of our quantization process.
> >
> > [1] Quarot: Outlier-free 4-bit inference in rotated llms, arXiv 2024.
>
>
> **W3**： Massive outliers and attention sinks
> > - Thank you for the clarification. We will correct the distinction between massive activations and attention sinks in the revised manuscript.
>
>
> **Q1**: DuQuant on Mistral and Phi models.
>
> > - We appreciate the inquiry and have extended the application of DuQuant to include Mistral and Phi models under 4-bit WA quantization. The PPL results are shown in the table below:
> >
> >   |Mistral-7B|WiKi| C4|
> >   |-|-|-|
> >   |FP16| 5.25| 7.75|
> >   |RTN| 306.26| 300.07|
> >   |SmoothQuant |100.59|158.02|
> >   |OmniQuant|5490.31|6094.82|
> >   |Atom| 8.65|12.43|
> >   |DuQuant| **5.86**| **8.48**|
> >
> >   |Phi2-2.8B|WiKi|C4|
> >   |-|-|-|
> >   |FP16 |9.71|12.76|
> >   |RTN|230.59|253.79|
> >   |SmoothQuant| 63.84|83.24|
> >   |OmniQuant|NaN|NaN|
> >   |Atom| 35.72|41.26|
> >   |DuQuant|**20.65**|**22.49**|
> >
> > - From the table, we can observe that DuQuant **largely surpasses other baselines**, particularly with **Mistral-7B**. Regarding the Phi2-2.8B model, it often experiences **instability** in matrix multiplication **between queries and values**, leading to **overflow** issues and posing great challenges to quantization. However, while DuQuant may not perform as well as FP models, it still significantly outperforms other baselines.
> > - In addition, we have visualized the massive outliers in the **down_proj** layer of the **Mistral-7B** model and the feature space after our dual transformations. These visualizations are available in the PDF file included in the general response section. It can be observed that our DuQuant perfectly eliminates these outliers.
> > - These results underscore the effectiveness of our dual transformation approach in addressing massive outliers **across various types of LLMs**.

---

> ### Comment · Reviewer_Rbrg · 2024-08-07
> **Reply to Rebuttal**
>
> Thank you for the detailed response to the reviews. After reading all reviews and responses as well as the global response, I will keep my score. I would have liked to see more results on long form generation (both memory, which was provided, as well as speedup and accuracy), but overall it's a strong work.

---

> > ### Author Response · Authors · 2024-08-13
> > **Speedup for Decoding Stage and Results for Long-context Generation**
> >
> > > - We appreciate the reviewer's detailed feedback, which is crucial for improving our work. The current generation of LLMs usually splits into pre-filling and decoding phases and deploys on two separate machines [1]. As we have already provided speedup/memory usage results for the pre-filling stage in the previous response and original paper, we further measure the **decoding stage speedup**.
> > >
> > > - In the decoding stage, batching the token generation phase yields high throughput without any downside [1]. Consequently, we enlarge the batch size to 64 in the decoding stage and measure speedup along with memory usage for one LLaMA2-7B layer, constrained by the 24 GB memory of an RTX 3090. We set the pre-filling sequence length at 2048 and decode for 128 steps. The results are presented below:
> > >
> > >   | One Layer, INT4, BS=64 | Time (ms) | Saving Factor | Memory (GB) | Saving Factor |
> > >   | ---------------------- | --------- | ------------- | ----------- | ------------- |
> > >   | FP16                   | 659       | -             | 3.550x       | -             |
> > >   | SmoothQuant            | 437       | 1.508x         | 1.669       | 2.127x         |
> > >   | QLLM                   | OOM       | -             | OOM         | -             |
> > >   | QuaRot                 | 457       | 1.442x         | 1.678       | 2.116x         |
> > >     | DuQuant                | 499       | 1.321x         | 1.677       | 2.117x         |
> > >
> > >     - From the table, the results demonstrate that DuQuant maintains speedup and memory usage comparable to QuaRot while delivering superior performance.
> > >
> > > - To further enhance real-world application speedup, we are grateful for the reviewer’s suggestion and committed to (1) developing more advanced W4A4 kernels to enhance decoding speedup in future work, or (2) combining our methods, which are compatible, with other decoding speedup techniques, such as speculative decoding, to substantially improve the overall speedup of DuQuant.
> > >
> > > - Responding to your interest in long-term generation results, we have included additional evaluations with LongBench, designed for long-context scenarios. With a **maximum generation length of 3500**, DuQuant significantly outperforms other baselines. We list the average results for Vicuna models as follows, while for the more detailed results please refer to the response to Reviewer XTv7 W1.
> > >
> > >   | Vicuna      | 7B (Avg.) | 13B (Avg.) |
> > >    | ----------- | --------- | ---------- |
> > >   | FP16        | 39.21     | 40.77      |
> > >    | SmoothQuant | 4.62      | 2.73       |
> > >   | OmniQuant   | 1.56      | 3.93       |
> > >    | Atom        | 33.19     | 30.61      |
> > >   | DuQuant     | **37.25** | **38.75**  |
> > >
> > > [1] Patel, Pratyush, et al. "Splitwise: Efficient generative llm inference using phase splitting." *2024 ACM/IEEE 51st Annual International Symposium on Computer Architecture (ISCA)*. IEEE, 2024.

---

> ### Comment · Reviewer_Rbrg · 2024-08-13
> **Response to Long-Context Results**
>
> These results look very reasonable. Because they resolve my concern about the long-context properties of DuQuant compared to other methods, I will raise my score. Please add these results to the main paper. Along with that, I would recommend to try the method on as many Llama-class models as possible (my understanding is that you don't need to rewrite the kernels for this), beyond just Vicuna. The results for Llama 3.1 8B or 70B (if hardware allows) would probably be the most practically relevant.

---

> > ### Author Response · Authors · 2024-08-13
> > **To Reviewer Rbrg**
> >
> > We are pleased to have addressed all concerns regarding the long-context properties of DuQuant. We appreciate the reviewer's decision to raise the score and will ensure to include these results in the main paper. Additionally, we plan to extend our experiments to LLaMA 3.1 series models to further validate the effectiveness of DuQuant. Again, we thank the reviewer for these invaluable suggestions and look forward to further enhancing our work with these additional evaluations.

---

### Official Review · Reviewer_8e4p · 2024-07-03

**Soundness:** 3
**Presentation:** 2
**Contribution:** 2
**Rating:** 7
**Confidence:** 4

**Summary:**

This work proposed a transformation (composition of orthogonal and permutation transformation) that makes LLMs more quantization-friendly (accounting for the presence of outlier features). The approach is validated on several modern LLMs from Llama-1,2,3 families.

**Strengths:**

The introduced method makes sense and targets the specific case of Massive outliers, that is not accounted in previous weight+activation approaches. The obtained results are pretty strong and achieve state-of-the-art at W4A4 quantization.

This approach is pretty fast and takes a couple of minutes even on large LLMs.

This work conducts a study on location and impact of specific types of outliers.

Speedups are quite significant.

**Weaknesses:**

The idea of applying rotation transformation for simplifying weight/activation quantization is not novel and (to my knowledge) was first introduced in QuaRot. The introduced method proposes a specific form of orthogonal/permutation matrices. QuaRot results for Llama-2-7b quantization reported in QuaRot paper are significantly better than the one presented in Table 7 and only marginally inferior to DuQuant numbers. I would suggest comparing with the numbers from original paper for fairness.

**Questions:**

* Do the learned rotation possess incoherence property [1]?

* Can one boost GPTQ performance further (as it is strictly stronger method compared to RTN) with DuQuant rotation matrices?

* What is the context length for Llama-3-8B evaluation? Typically, one uses the training context length for comparison (8k for Llama-3 model family).

[1] Chee, Jerry, et al. "Quip: 2-bit quantization of large language models with guarantees." Advances in Neural Information Processing Systems 36 (2024).

**Limitations:**

See Weaknesses.

---

> ### Author Rebuttal · Authors · 2024-08-06
>
> We greatly appreciate the reviewer's constructive comments on our paper. We will respond to the reviewer's feedback with detailed explanations for each point.
>
> **W1**: Highlight of Ours and detailed comparison with QuaRot.
>
> >   - We appreciate the reviewer's comments. We have dedicated **Appendix F** to highlighting our novel contributions and demonstrating our superiority over QuaRot. We summarize our key contributions below.
> >   - Instead of adopting the Hadamard rotation used in QuaRot, we use a greedy search algorithm that leverages prior knowledge to compute an **approximately ideal rotation matrix** that (1) is orthogonal and (2) specifically targets the positions of outliers, redistributing them across adjacent channels. **Figure 4** advocates the superiority of DuQuant in mitigating outliers compared to the Hadamard rotation.
> >   - We introduce the **zigzag permutation** that reduces activation magnitude variance between blocks, further reducing the overall outliers. The ablation study in **Table 5** highlights the significance of this permutation.
> >   - Our rotation and permutation matrices **simultaneously smooth weights** and activations. Consequently, DuQuant avoids the time-consuming GPTQ used by QuaRot. **Table F17** illustrates the significantly higher efficiency of DuQuant.
> >   - Due to these contributions, DuQuant consistently outperforms QuaRot. Although the results reported in QuaRot are higher than our reproduced ones, **the experimental settings in the original paper of QuaRot differ from ours**.
> >     - We summarize these setting differences in the following table.
> >
> >       |Setting|Weight|Activation|Key/Value|Query|
> >       |-|-|-|-|-|
> >       |QuaRot|symmetric|symmetric|group-wise asymmetric|FP16|
> >       |DuQuant|asymmetric|asymmetric|asymmetric|asymmetric|
> >
> >   - The comparison reported in our paper is **fair**, as we reproduced QuaRot under our settings. The results in Table 7 and Table F15-F17 consistently demonstrate the superiority of DuQuant over QuaRot.
> >
> >   - **Under the setting utilized in the original paper of QuaRot**, as suggested by the reviewer, we also provide the results of DuQuant in the table below. **DuQuant still outperforms QuaRot** by a large margin in PPL and QA tasks. Note that all the results of QuaRot are directly brought from its original paper.
> >
> >     |LLaMA2 W4A4|Method|WiKi↓|C4↓|PQ↑|WG↑|HS↑|A-e↑|A-c↑|LA↑|Avg↑|
> >     |-|-|-|-|-|-|-|-|-|-|-|
> >     |7B QuaRot Setting|FP16|5.47|6.97|79.11|69.06|75.99|74.58|46.25|73.90|69.82|
> >     ||QuaRot-RTN|8.37|-|72.09|60.69|65.40|58.88|35.24|57.27|58.26|
> >     ||QuaRot-GPTQ|6.10|-|76.77|63.77|72.16|69.87|40.87|70.39|65.64|
> >     ||DuQuant|6.23|7.91|76.28|66.93|72.96|69.99|40.53|69.61|66.05|
> >     ||DuQuant-LWC|**6.01**|**7.67**|77.64|67.80|72.97|70.37|41.81|69.53|**66.69**|
> >     |13B QuaRot Setting|FP16|4.88|6.46|80.47|72.22|79.39 |77.48|49.23|76.75|72.59|
> >     ||QuaRot-RTN|6.09|-|77.37|67.32|73.11|70.83|43.69|70.66|67.16|
> >     ||QuaRot-GPTQ|5.40|-|78.89|70.24|76.37| 72.98|46.59|73.67|69.79|
> >     ||DuQuant| 5.39 |7.05|78.51|70.88|76.80| 74.62|48.21|73.92|**70.49**|
> >     ||DuQuant-LWC|**5.27**|**6.93**|78.73|70.88| 77.20|74.07|47.27| 73.96 |70.35|
>
> **Q1**: Does the rotation transformation possess an incoherence property [1]?
> > Yes, our learned rotation transformation indeed possesses the incoherence property [1].
> >    - As described in [1], the incoherence of weight and Hessian matrices is ensured by multiplying them with a Kronecker product of **random orthogonal matrices**.
> >    - While our approach includes a greedy search step to learn the matrix, the final matrix $\hat{\mathbf{R}}$ obtained remains **orthogonal**. This is because the product of orthogonal matrices $\hat{\mathbf{R}} = \mathbf{R}^1\mathbf{R}^2\cdots \mathbf{R}^n$ remains orthogonal.
> >
> > [1] Quip: 2-bit quantization of large language models with guarantees, NeurIPS 2024.
>
> **Q2**: Can one boost GPTQ performance further with DuQuant rotation matrices?
> > We appreciate this valuable question. We demonstrate below that DuQuant is **compatible with and contributory to stronger methods, including GPTQ.**
> >    - We implement DuQuant+GPTQ by applying GPTQ exclusively on the four key layers after a dual transformation so that the computational overhead introduced by GPTQ is minimized. These four key layers are selected according to the compression difficulty, as suggested in ShortGPT [2].
> >    - The table below shows that this combination leads to an additional performance boost, further validating the effectiveness of DuQuant.
> >
> >       |LLaMA2 W4A4|WiKi|C4|
> >       |-|-|-|
> >       |7B-DuQuant|6.28|7.90|
> >       |7B-DuQuant+GPTQ|6.15|7.73|
> >       |13B-DuQuant|5.42|7.05|
> >       |13B-DuQuant+GPTQ|5.39|6.96|
> >
> > [2] Shortgpt: Layers in large language models are more redundant than you expect, arXiv 2024.
>
> **Q3**: Context length for LLaMA3-8B evaluation.
> > - We follow OmniQuant [3] and [4] to set the context length to 2048 in Table 4 of our paper. We will include this detail in Appendix C.
> > - We also follow the reviewer's suggestion to conduct a PPL evaluation for all baselines under an 8k context length and 4-bit quantization, as shown below:
> >
> >   |LLaMA3-8B (8k)|WiKi|C4|PTB|
> >   |-|-|-|-|
> >   |FP16|6.14|8.62|9.91|
> >   |SmoothQuant|225.65|242.70|277.38|
> >   |Atom|18.07|26.76|34.97|
> >   |DuQuant|7.57|12.24|12.44|
> >
> >  - Despite the increase in context length, all methods perform better. This improvement is attributed to 1) the 8k context length ensures the model is evaluated under the same conditions as it was trained, 2) the longer context length provides more historical context for the model to predict the next word, which helps the model generate text more accurately.
> >  - It can be observed that DuQuant continues to outperform the baselines.
> >
> > [3] Omniquant: Omnidirectionally calibrated quantization for large language models, ICLR 2024.
> >
> > [4] How good are low-bit quantized llama3 models? an empirical study, arXiv 2024.

---

> ### Comment · Reviewer_8e4p · 2024-08-07
>
> Thanks for your detailed response and clarifications. My concerns regarding the fairness of the evaluation setup are resolved. Hence, I have decided to raise the score.

---

> > ### Author Response · Authors · 2024-08-12
> > **To Reviewer 8e4p**
> >
> > We are delighted to see that the major concerns raised by the reviewer have been successfully addressed. We would like to express our deep appreciation for the reviewer's dedicated time and effort in scrutinizing our paper and providing invaluable feedback.

---

### Official Review · Reviewer_XTv7 · 2024-07-11

**Soundness:** 3
**Presentation:** 3
**Contribution:** 3
**Rating:** 8
**Confidence:** 4

**Summary:**

The paper presents a new post-quantization method (DuQuant) that targets low-precision (4-bit / 6-bit) weight and activation quantization. The authors show how the presence of massive outliers affects quantization when using existing methods (smoothing is not sufficient with SmoothQuant / OmniQuant training is not stable for layers exhibiting massive outliers). They propose their method for better handling both normal and massive outliers - by utilizing orthogonal rotation and permutation matrices, with a simple zigzag permutation scheme for a better / more even distribution of outliers. They provide theoretical proofs for both their rotation and zigzag-permutation operations grounding their proposed algorithm. They showcase the strength of the propsed methods as it can be enabled with simple quantization, and not rely on expensive quantization methods like GPTQ to achieve new state of art quantized models across a range of different models.

**Strengths:**

1. The paper presents a theoretically grounded approach to low-precision (4-bit / 6-bit) quantization of LLMs.
2. The paper showcases how massive outliers are challenging for existing methods to adapt to (for e.g., SmoothQuant just fails and OmniQuant sees unstable gradients in training layers with massive outliers). The authors also particularly show that the down-projection layers in FFNs particularly face massive outliers - which inhibit effectively their quantization.
3. The authors propose a new RAP (rotation and permutation) based method for enabling more even distribution of outliers from the activations to weights.
    - They first start with the smoothing operation proposed in SmoothQuant, through ablations they show this helps get better post-quant model quality.
    - They follow this up with a rotation operation, with the constraints that it should be able to as evenly distribute the outliers through the matrix multiplications. They show that a single rotation process cannot handle this effectively and design an iterative but greedy process that solves this using block-diagonal rotation matrices.
    - Followed by this, they propose a new zigzag permutation operation to evenly distribute large outliers across different blocks. Finally, they apply another rotation to ensure that the outliers are maximally reduced and spread across the weights.
4. They follow the proposal of the method through theoretical analysis, showing how their methods will results in either optimal results / have bounds in each phase of the smoothening to ensure maximal quality.
5. They propose two variants of the algorithm - one standard with specific activation / weight clipping coeffs and another with LWC enabled from the OmniQuant algorithm. They follow these with 4-bit and 6-bit results for the LLaMA-1,2,3 and Vicuna-1.5 models, showcasing that their method outperforms other existing methods in these settings.
6. They provide a comparison with the recently proposed QuaRot method and how their proposed algorithm performs better than the algorithm, while being competitive for implementation performance.

**Weaknesses:**

1. Most of the evals (except MT-Bench) are logit-based evals (and not generative). This has a side effect of hiding some of the inherent limitations of low-precision quantization algorithms (i.e., error accumulation across generated tokens).
2. One limitation of the proposed benchmarks (for implementation performance) is that they measure the prefill performance, but do not show any generation performance. This typically dominates over prefill performance.
3. One thing is the reporting of results is not consistent across the models - for eg. some tables use Atom for reporting model quality at 4-bits, but some models do not report this performance. While the benefits of the method are clear given the higher accuracies on downstream tasks, it is difficult to judge how the differences translate on models of higher quality (e.g. Llama-2 vs Llama=3 70B).

**Questions:**

1. Many times the perplexity numbers and downstream performance are not 1:1 correlated. This most likely has to do with the standard error of the downstream tasks? If so, it will be good to clarify this for the reader.
2. Figure 5 is not clear in explaining the different settings - what do Perm 0, 1, and 2 correspond to? Also in the paragraph above the figure, in lines 312-313, it is not clear which Figure is being referred to.
3. For Table 6, it will be better to explain how to read the table. It took multiple passes to understand the full setup of the ablations and the associated results in the table.
4. It is surprising that the method is largely calibration free - indicating that the outliers are more a property of the model weights, and activations are suffering as a by-product of this? can the authors clarify their intuition around this.

**Limitations:**

The authors address the limitations and broader impacts of their work. One thing they do mention in the checklist is reporting of statistical signficance - which I do not see anywhere in the paper. Can authors point to where these results are? Or equivalently change the checklist to reflect this.

---

> ### Author Rebuttal · Authors · 2024-08-06
>
> We sincerely thank the reviewer for providing valuable feedback. We detail our response below point by point. Please kindly let us know whether you have any further concerns.
>
> **W1**: More evaluations on generative tasks.
> > - To better access the generative ability of quantized models, we evaluate DuQuant on LongBench and report INT4 results below:
> >
> >   |Vicuna|Setting|RepoBench-P|MultiFieldQA-en|GovReport|MultiNews|MultiFieldQA-zh|2WikiMultihopQA|
> >   |-|-|:-:|:-:|:-:|:-:|:-:|:-:|
> >   |7B|FP16|48.23|38.30|27.93|26.91|32.56|18.02|
> >   ||SmoothQuant|25.92|4.66|2.62|6.05|0.88 |2.02|
> >   ||OmniQuant|14.97|2.30|2.51|2.64|1.40|0.48|
> >   ||Atom|29.34|31.15|23.60|24.60|21.55|17.10|
> >   ||DuQuant|47.66|35.62|25.66|25.85|29.56|15.09|
> >   |13B|FP16|43.08|42.69|28.43|26.53|40.44|29.40|
> >   ||SmoothQuant|11.57|1.64|2.81|3.54|0.82|1.39|
> >   ||OmniQuant|8.46|4.32|0.74|2.83|1.06|0.75|
> >   ||Atom|37.31|37.31|19.34|23.39|28.02|15.16|
> >   ||DuQuant|38.09|44.12|26.97|26.59|30.85|22.07|
> >
> >   |Vicuna|Setting|TriviaQA|QMSum|LSHT|DuReader|NarrativeQA|Qasper|SAMSum|TREC|Avg|
> >   |-|-|:-:|:-:|:-:|:-:|:-:|:-:|:-:|:-:|:-:|
> >   |7B|FP16|82.59|21.07|22.25|25.53|14.96|23.27|41.06|66.00|39.21|
> >   ||SmoothQuant|1.62|2.00|0.00|4.24|1.75|4.11|1.55|15.00|4.62|
> >   ||OmniQuant|0.81|3.93|0.00|1.87|1.10|1.62|0.61|1.00|1.56|
> >   ||Atom|67.20|20.24|17.25|19.41|11.57|17.97|37.94|58.00|33.19|
> >   ||DuQuant|78.91|21.15|19.00|23.15|11.31|19.98|42.24|64.00|**37.25**|
> >   |13B|FP16|86.81|21.24|24.00|27.57|15.41|24.41|41.97|68.00|40.77|
> >   ||SmoothQuant|1.83|2.95|0.00|6.71|0.97|2.18|0.35|1.50|2.73|
> >   ||OmniQuant|1.13|1.78|0.00|13.83|0.62|0.68|0.45|9.00|3.93|
> >   ||Atom|80.75|20.23|21.00|21.79|8.81|17.67|38.72|59.00|30.61|
> >   ||DuQuant|83.04|20.72|23.75|26.02|13.36|18.93|42.67|66.50|**38.75**|
>
> **W2**: The generation stage performance.
>
> > - As the prefill phase is usually compute-bound while the decoding phase is known to be memory-bound [1], we compare the **memory consumption reduction** of DuQuant with other baselines during the generation stage. Evaluations were conducted on a 3090 with batch size 1.
> >
> >   |LLaMA2-7B, INT4|Memory (GB)|Saving Factor|
> >   |-|-|-|
> >   |FP16|13.638|-|
> >   |SmoothQuant|3.890| 3.506x|
> >   |QLLM|3.894|3.502x|
> >   |QuaRot|3.891|3.505x|
> >   |DuQuant|3.893|3.503x|
> >
> > [1] Quarot: Outlier-free 4-bit inference in rotated llms, arXiv 2024.
>
> **W3**: Baselines across all models.
> > We acknowledge the omission of some baseline results for the LLaMA2-70B and LLaMA3-70 models. This is because,
> >   - We encountered NaN perplexity results on the LLaMA2-70B and LLaMA3-70B models for some baselines, like Atom, leading us to exclude these results from QA task evaluations.
> >   - Possibly due to inadequate management of massive outliers, AffineQuant and OmniQuant experienced **instability** when learning on the 70B models, often resulting in gradient explosions.
>
> **Q1**: The discrepancies between PPL and other downstream tasks.
> > - PPL is utilized to assess the generation abilities of LLMs, while downstream tasks like QA in our paper mainly evaluate the comprehension abilities of LLMs. They focus on different aspects of model capacities, which may result in discrepancies between tasks. In addition, PPL might not be a reliable evaluation to reflect the model’s effectiveness in real-world tasks [2]. Thus, to better evaluate DuQuant under practical applications, we experiment on LongBench as your suggestion in W1.
> >
> > [2] Longbench: A bilingual, multitask benchmark for long context understanding, ACL 2024.
>
> **Q2 & 3**: Detailed illustrations of Figure 5 and Table 6.
> > - We apologize for any confusion caused by the unclear descriptions and will clarify these illustrations in the revised paper.
> > - **Figure 6**: This figure shows ablations of rotation and permutation frequencies in DuQuant. "Perm 0" indicates a single rotation, "Perm 1" signifies two rotations with one channel permutation, and "Perm 2" includes three rotations with two permutations. Results show that "Perm1" offers the best balance between PPL and inference speed, which we adopted as the final configuration in DuQuant.
> > - **Table 5**: The table presents ablations on four distinct operations within DuQuant. A check mark indicates the inclusion of an operation. The configurations tested are 1) only the smoothing technology like SmoothQuant; 2) one rotation following the smoothing operation; 3) a sequence of rotation, permutation, and another rotation without smoothing; and 4) the full DuQuant approach. These results underscore the contribution of each component to the overall effectiveness of DuQuant.
>
> **Q4**: Discussion about calibration-free experiments.
> > - Our findings in Appendix E.4 demonstrate that DuQuant does not depend on specific calibration data, suggesting that outliers are inherent to certain model layers and are **characteristic of the model weights or modules**. This is supported by two recent works:
> >   -  [3] identifies consistent massive outliers specifically at the FFN down projection layer in GLU-based LLMs, such as LLaMA, Mistral, Mixtral, SOLAR, and Gemma.
> >   -  [4] investigates the impact of calibration sets on quantization, finding that while OPT models are sensitive to varying calibration sets, newer models like Llama, Command-R, and Mistral show robustness to outliers and stable activations.
> > - These insights confirm that outliers exhibit **consistent distributions** for recent LLMs, which is a property of the weights and modules.
> >  [3] Mitigating Quantization Errors Due to Activation Spikes in GLU-Based LLMs, arXiv 2024.
> >
> >  [4] Outliers and Calibration Sets have Diminishing Effect on Quantization of Modern LLMs, arXiv 2024.
>
> **L1**: Statistical significance.
> > - We apologize for the oversight. Our study used a fixed seed for all quantization operations, following standards in post-training quantization, and thus did not report statistical significance. We will correct this in the checklist of our revised manuscript.

---

> > ### Comment · Reviewer_XTv7 · 2024-08-07
> >
> > Thank you for the detailed response to the reviews and additional experimental results. After reading all reviews and response, and the overall global response, I will retain my score.
> >
> > I encourage the authors to do the following for the camera ready version:
> >     1. Add the generative evals (W1 response) and memory profile (W2 response) to the supplementary and link them in the main work for readers.
> >     2. Add the clarifications for Q 2 & 3 to the paper for easier reading
> >     3. Address the Q4 answers in the supplementary work.

---

> > > ### Author Response · Authors · 2024-08-12
> > > **To Reviewer XTv7**
> > >
> > > We are pleased that the concerns raised by the reviewer have been addressed, and we will incorporate the additional experimental results and clarifications during our discussion into the revised version. Thanks again for the time and effort the reviewer has dedicated to reviewing our paper and providing valuable feedback.

---

### Official Review · Reviewer_nwuk · 2024-07-14

**Soundness:** 3
**Presentation:** 2
**Contribution:** 3
**Rating:** 8
**Confidence:** 3

**Summary:**

The paper explores new approaches in LLM quantization. The work tries to address the performance degradation due to massive outliers in the weights. The work show competitive performance across different settings, up to 4-bit weight activation quantization. The work also provides solid experiments and visualization on the effectiveness of the proposed methods.

**Strengths:**

1. I think the paper is well-written. The experiments, plots, and tables are well clearly tied to the story on large/abnormal outlier in the activation.
2. The work is easy to follow, especially in the methodology.
3. The results in the tables show that work offer very competitive performance comparing to previous methods.

**Weaknesses:**

1. I am a bit puzzled by the results in Table.1 vs Table.2. It is shown that lower perplexity does not necessarily reflect in higher benchmark accuracy. This conflict exists within DuQuant (DuQuant vs DuQuant_LWC) and other methods like AffineQuant,OmniQuant vs QLLM. The authors might want to consider adding some justification to this conflict to further improve the soundness of the tables.

**Questions:**

From the benchmarks, it is shown that the proposed method almost matches the fp16 results (unquantized). It would be interesting to see the mt-bench of the quantized vs fp16 model to see if such an observation still holds.

---

> ### Author Rebuttal · Authors · 2024-08-06
>
> **W1**: Confusion Regarding the Discrepancies Between Table 1 and Table 2 Results.
>
> > - We would like to provide individual clarifications for the results in Table 1 and Table 2 below and explain the reasons behind their discrepancies.
> > - **Table 1** presents perplexity (PPL) results on the WikiText-2 and C4 datasets, which are typically used to evaluate the language generation capabilities of LLMs. **Table 2** shows the results of the Common Sense QA task, which focuses on the model's comprehension abilities, different from the generation abilities PPL focused on. As Table 1 and Table 2 **emphasize different aspects of model performance**, this explains the conflicting results between the two tables.
> > - In addition, PPL usually reflects how well a model predicts a sequence of words, and it **cannot** measure the model's effectiveness in handling sequence-level tasks in practical applications [1, 2]. To further evaluate our model's generative capabilities, we have added a comprehensive comparison of DuQuant against other state-of-the-art baselines on the **LongBench** [1], which includes a variety of **generative tasks** to provide a broader evaluation. The W4A4 results are presented as follows:
> >
> >   |Vicuna|Setting|RepoBench-P|MultiFieldQA-en|GovReport|MultiNews|MultiFieldQA-zh|2WikiMultihopQA|
> >   |-|-|:-:|:-:|:-:|:-:|:-:|:-:|
> >   |7B|FP16|48.23|38.30|27.93|26.91|32.56|18.02|
> >   ||SmoothQuant|25.92|4.66|2.62|6.05|0.88 |2.02|
> >   ||OmniQuant|14.97|2.30|2.51|2.64|1.40|0.48|
> >   ||Atom|29.34|31.15|23.60|24.60|21.55|17.10|
> >   ||DuQuant|47.66|35.62|25.66|25.85|29.56|15.09|
> >   |13B|FP16|43.08|42.69|28.43|26.53|40.44|29.40|
> >   ||SmoothQuant|11.57|1.64|2.81|3.54|0.82|1.39|
> >   ||OmniQuant|8.46|4.32|0.74|2.83|1.06|0.75|
> >   ||Atom|37.31|37.31|19.34|23.39|28.02|15.16|
> >   ||DuQuant|38.09|44.12|26.97|26.59|30.85|22.07|
> >
> >   |Vicuna|Setting|TriviaQA|QMSum|LSHT|DuReader|NarrativeQA|Qasper|SAMSum|TREC|Avg|
> >   |-|-|:-:|:-:|:-:|:-:|:-:|:-:|:-:|:-:|:-:|
> >   |7B|FP16|82.59|21.07|22.25|25.53|14.96|23.27|41.06|66.00|39.21|
> >   ||SmoothQuant|1.62|2.00|0.00|4.24|1.75|4.11|1.55|15.00|4.62|
> >   ||OmniQuant|0.81|3.93|0.00|1.87|1.10|1.62|0.61|1.00|1.56|
> >   ||Atom|67.20|20.24|17.25|19.41|11.57|17.97|37.94|58.00|33.19|
> >   ||DuQuant|78.91|21.15|19.00|23.15|11.31|19.98|42.24|64.00|**37.25**|
> >   |13B|FP16|86.81|21.24|24.00|27.57|15.41|24.41|41.97|68.00|40.77|
> >   ||SmoothQuant|1.83|2.95|0.00|6.71|0.97|2.18|0.35|1.50|2.73|
> >   ||OmniQuant|1.13|1.78|0.00|13.83|0.62|0.68|0.45|9.00|3.93|
> >   ||Atom|80.75|20.23|21.00|21.79|8.81|17.67|38.72|59.00|30.61|
> >   ||DuQuant|83.04|20.72|23.75|26.02|13.36|18.93|42.67|66.50|**38.75**|
> >  - From the table, our DuQuant outperforms other baselines by a clear margin, representing the superior ability for long context generation tasks.
> >
> > [1] Longbench: A bilingual, multitask benchmark for long context understanding, ACL 2024.
> >
> > [2] Do long-range language models actually use long-range context? EMNLP 2021.
>
> **Q1**: MT-Bench evaluation between DuQuant and FP16 models.
> > - As suggested, we conducted additional comparisons using the MT-Bench between our INT4 quantized models and the FP16 models. The results are presented in the table below.
> >
> > | DuQuant vs FP16 | Former Win | Tie  | Former Loss |
> > | --------------- | ---------- | ---- | ----------- |
> > | Vicuna-7B       | 36         | 56   | 68          |
> > | Vicuna-13B      | 43         | 53   | 64          |
> >
> > - The results indicate that our **quantized models perform comparably to FP16**, underscoring the effectiveness of our dual transformation approach in maintaining high accuracy even with reduced precision.

---

> > ### Comment · Reviewer_nwuk · 2024-08-10
> >
> > I thank the authors for the rebuttal and clarification. I think the rebuttal addresses my concern and I will raise the score.

---

> > > ### Author Response · Authors · 2024-08-12
> > > **To Reviewer nwuk**
> > >
> > > We thank the reviewer for acknowledging our efforts to address the concerns. We are grateful for the decision to reconsider the score based on our responses. The constructive feedback has greatly enhanced our manuscript.

---

### Official Review · Reviewer_x516 · 2024-07-15

**Soundness:** 4
**Presentation:** 3
**Contribution:** 3
**Rating:** 7
**Confidence:** 3

**Summary:**

The paper proposes a new LLM quantization method named DuQuant (for Dual transformations Quantization). This method is able to quantize the weights and activations of an LLM to 4 or 6 bits without losing significant precision.

The paper identifies the issue of Massive outliers in the activations of a LLM. These are outliers within the outliers, having magnitudes in the order of 100-1000. Traditional quantization methods, such as SmoothQuant, cannot deal with these outliers, because they attempt to address the errors locally. DuQuant, on the other hand, uses a sequence of rotation, permutation, and another rotation to spread these outliers out across many weights, resulting in a more accurate capture.

The results are supported with theoretical results as well as experiments. DuQuant outperforms the baselines on perplexity, QA and MT-bench.

**Strengths:**

The paper is well-written and easy to understand (apart from a few minor typos). I especially like figure 2 that give a strong intuition to how the method works and also why it eliminates Massive outliers. The algorithm, the justification and the results are easy to follow.

The experimental results are very convincing. They demonstrate an improvement in prediction quality in W4A4 across the board. The paper shows extensive experiments looking at various Llama models as well as Vicuna. Compared to the baselines (SmoothQuant, AffineQuant, OmniQuant, Atom, QLLM), DuQuant always performs the closest to the uncompressed model. This result is shown across various tasks: perplexity, QA and MT bench.

The paper also includes ablations and runtime costs of the required transforms: the transforms incur an additional 8.9-9.3% computational cost at inference time compared to standard W4A4.

**Weaknesses:**

A shortcoming of the paper is the lack of close comparison to QuaRot. QuaRot is a very similar technique and it deserves to be in the related works section as well as be a baseline in the experiments. While there is a detailed comparison in the appendix, the work should also be mentioned in the main body.

The paper could be improved at places, but I don't consider any of these shortcomings a major issue:
* The paper has numerous typos (eg.: Line 11: Typo, Line 188: typo, Line 313: Figure reference missing). Please proofread.
* The paper is extremely dense, its sometimes difficult to distinguish the figure captions from the main body. Please use proper paddings for tables and figures where possible.
* The description of how the rotation matrix is constructed could be more detailed (Lines 141 - 147). Given that this is a core algorithmic contribution of the paper, it should be clear to the reader how it works. Perhaps a figure or pseudocode would be helpful here.

**Questions:**

No outstanding questions.

**Limitations:**

No limitations.

---

> ### Author Rebuttal · Authors · 2024-08-06
>
> Thanks for your time in dealing with our work. We will answer the question and discuss point by point as follows. We hope that our response satisfactorily addresses the issues you raised. Please feel free to let us know if you have any additional concerns or questions.
>
> **W1**: The comparison with the QuaRot part should be highlighted in the main text.
>
> > - Thanks for the suggestion. We acknowledge that QuaRot is an important baseline. As suggested, we will move some of the comparison experiments from the appendix to the main text and highlight them in the related work section.
> >
> > - To further compare with QuaRot, we implemented DuQuant under **QuaRot original quantization settings**, which is different from the 4-bit per-token activation quantization and per-channel weight quantization setting used in our paper. We compared DuQuant with **the original results reported in QuaRot paper**. It can be observed from the following table that DuQuant still surpasses QuaRot on both PPL and QA evaluations.
> >
> >   | LLaMA2 W4A4        | Method      | WiKi↓    | C4↓      | PQ↑   | WG↑   | HS↑   | A-e↑  | A-c↑  | LA↑   | Avg↑      |
> >   | ------------------ | ----------- | -------- | -------- | ----- | ----- | ----- | ----- | ----- | ----- | --------- |
> >   | 7B QuaRot Setting  | FP16        | 5.47     | 6.97     | 79.11 | 69.06 | 75.99 | 74.58 | 46.25 | 73.90 | 69.82     |
> >   |                    | QuaRot-RTN  | 8.37     | -        | 72.09 | 60.69 | 65.40 | 58.88 | 35.24 | 57.27 | 58.26     |
> >   |                    | QuaRot-GPTQ | 6.10     | -        | 76.77 | 63.77 | 72.16 | 69.87 | 40.87 | 70.39 | 65.64     |
> >   |                    | DuQuant     | 6.23     | 7.91     | 76.28 | 66.93 | 72.96 | 69.99 | 40.53 | 69.61 | 66.05     |
> >   |                    | DuQuant-LWC | **6.01** | **7.67** | 77.64 | 67.80 | 72.97 | 70.37 | 41.81 | 69.53 | **66.69** |
> >   | 13B QuaRot Setting | FP16        | 4.88     | 6.46     | 80.47 | 72.22 | 79.39 | 77.48 | 49.23 | 76.75 | 72.59     |
> >   |                    | QuaRot-RTN  | 6.09     | -        | 77.37 | 67.32 | 73.11 | 70.83 | 43.69 | 70.66 | 67.16     |
> >   |                    | QuaRot-GPTQ | 5.40     | -        | 78.89 | 70.24 | 76.37 | 72.98 | 46.59 | 73.67 | 69.79     |
> >   |                    | DuQuant     | 5.39     | 7.05     | 78.51 | 70.88 | 76.80 | 74.62 | 48.21 | 73.92 | **70.49** |
> >   |                    | DuQuant-LWC | **5.27** | **6.93** | 78.73 | 70.88 | 77.20 | 74.07 | 47.27 | 73.96 | 70.35     |
>
> **W2**: Typos and article typesetting. Detailed description of how to construct rotation matrix.
>
> > - Thank you for your detailed feedback. We will diligently correct the typos and enhance the layout in our revised manuscript to improve its readability.
> > - Regarding the construction of the rotation matrix, we have provided a detailed pseudo code below and will involve this part in the revised manuscript to elucidate the process more clearly for the readers.
> >
> > ```
> >   INPUT: pre-initialized rotation matrix R0, greedy search steps n, activation matrix X with shape of [N, C]
> >   OUTPUT: rotation matrix R
> >   FUNCTION get_rotation_matrix(X, R0, n)
> >   	R = eye(C) # size: [C, C]
> >   	for i in 1...n: # greedy search loop
> >   		channel_max = X.abs().max(dim=0).values # size: [C]
> >   		outlier_channel = argmax(channel_max)
> >
> >   		Obtain randomly initialized orthogonal matrix Q' with the shape of [C-1, C-1]
> >   		Q' = concat([zeros(n-1, 1), Q'], dim=1)
> >   		Q = concat([zeros(1, n), Q'], dim=0)
> >   		Q[0, 0] = 1
> >   		R' = matmul(R0, Q)
> >
> >   		R'[:, outlier_channel], R'[:, 0] = R'[:, 0], R'[:, outlier_channel] # swap columns
> >   		R'[outlier_channel, :], R'[0, :] = R'[0, :], R'[outlier_channel] # swap rows
> >   		R = matmul(R, R')
> >   		X = matmul(X, R')
> >   	return R
> > ```

---

> > ### Comment · Reviewer_x516 · 2024-08-12
> > **Thank you for your reply**
> >
> > Thank you for the detailed reply and for addressing my concerns. I think the construction of the rotation matrix will be an important clarification in the revised manuscript.

---

> > > ### Author Response · Authors · 2024-08-13
> > > **To Reviewer x516**
> > >
> > > We are pleased that we have effectively resolved the key issues identified by the reviewer. We extend our sincere thanks for the reviewer’s thorough examination of our manuscript and the constructive feedback provided. We will revise the manuscript to include the details about the construction of the rotation matrix.

---

### Author Rebuttal · Authors · 2024-08-07

### **General Response for All Reviewers**

>**Summary**:
>
> We sincerely thank all reviewers for their valuable time and insightful feedback, which is very helpful in further improving the quality of our paper. We are grateful that the reviewers appreciate (1) "the technical contributions of dual transformations are notable" (XTv7, Rbrg); (2) "the method is well-motivated for managing massive/normal outliers" (all reviewers) ; (3) "the paper is well-written/well-organized" (x516, nwuk); and "the theoretical analysis guarantees the performance" (x516, XTv7, Rbrg). We are also encouraged that the experiments are "comprehensive" (YKYC) and the experimental results are "competitive" (x516) and "convincing" (nwuk, Rbrg). In addition, our speedup and runtime test are acknowledged by reviewers as "significant" (8e4p).

>
>
>In order to provide greater clarity on the revisions made to our paper and the experiments we conducted to address the reviewers' questions, we have summarized the modifications and experiments made during the rebuttal period as follows:


>**Additional Experiments**:
>
>- We provide additional experiments on **LongBench** to evaluate the generative ability of quantized models. (Reviewer XTv7 W1)
>- We apply DuQuant on **Mistral-8B** and **Phi2-2.8B** to demonstrate the effectiveness on other model types. (Reviewer Rbrg Q1)
>- We compare the DuQuant quantized Vicuna with the **FP16 models** on **MT-Bench**. (Reviewer nwuk Q1)
>- We implement DuQuant under **QuaRot quantization settings** and compare with results from their **original paper**. (Reviewer x516 Q1, Reviewer 8e4p W1)
>- We combine DuQuant with **GPTQ** to further boost the quantized model. (Reviewer 8e4p Q2)
>- The analysis of the **context length** for LLaMA3-8B evaluation. (Reviewer 8e4p Q3)
>- The analysis of the **memory usage** during the **decoding** stage. (Reviewer XTv7 W2)
>- The **comparison with baselines** for memory consumption reduction. (Reviewer Rbrg W1)
>- The analysis of **memory consumption, time usage, and model performance**. (Reviewer Rbrg W2)

>**Clarifications**:
>
>- The detailed comparison with QuaRot includes analysis from various perspectives (Reviewer 8e4p W1) and additional experiments. (Reviewer x516 Q1, Reviewer 8e4p W1)
>- The illustration of the discrepancies between PPL and other downstream tasks. (Reviewer nwuk W1, Reviewer XTv7 Q4)
>- The pseudo-code to clarify the construction of the rotation matrix. (Reviewer x516 W2)
>- The reason why the reporting of results is not consistent across the models. (Reviewer XTv7 W3)
>- The detailed illustrations of Figure 5 and Table 6. (Reviewer XTv7 Q5)
>- The discussion about calibration-free experiments. (Reviewer XTv7 Q6)

>**Attachment**:
>- We visualize the massive activations change with our DuQuant on Mistral-7B in the attached PDF.

---

### Decision · Program_Chairs · 2024-09-25

**Decision:**

Accept (oral)

**Comment:**

The authors propose a new method for weight and activation quantization in LLMs that targets the existing issue of large-magnitude activations and their effect on quantization. Their method applies a carefully-designed linear transformation to better distribute these outliers, with theoretical and practical demonstrations of success. Initial reviews of the paper were positive, and increased further after clarifications and additional results during the rebuttal and discussion phase. I concur with the reviewers and recommend acceptance. The authors should carefully incorporate feedback from the reviewers into the final revision.